# Origin of slow earthquake statistics in low-friction soft granular shear

Yuto Sasaki ✉ & Hiroaki Katsuragi ✉

Slow earthquakes differ from regular earthquakes in their slower moment release and size distribution dominated by smaller events. However, the physical origin of these slow earthquake statistics remains controversial. In this work, we experimentally demonstrate that their characteristics emerge from low-friction soft granular shear. To model slow-earthquake fault materials under hydrothermal conditions, we use a low-friction soft hydrogel particle layer floating on lubricating fluid and conduct stick-slip experiments. The observed slip events follow the same laws of both moment release rate and size distribution as with slow earthquakes, contrasting with frictional rigid granular shear. Slip size is determined by the competing effects of shear localization and pressure enhancement with decreasing porosity. These findings indicate that low friction and particle softness in sheared granular systems with sparse contact structures cause slow earthquake statistics, which may be driven by pore fluid dynamics and shear localization within hazardous fault zones.

Slow earthquakes are distinctively slower slip phenomena than regular earthquakes of equivalent size[1,2]. Between slow and regular earthquakes, the statistics of seismic moment release rate[2] and moment size distribution[3] often exhibit different characteristics. While the moment rate of regular earthquakes follows the cubic moment-duration scaling $M_0 \propto T^3$ with seismic moment $M_0$ and duration $T$ (refs. 4,5), that of slow earthquakes seems to have a constant upper bound as $M_0 \propto T$ (ref. 2). Moreover, while regular earthquakes exhibit a size distribution following a power law[6,7], slow earthquakes frequently obey exponential[3,8,9] or power-law distributions with a higher exponent[10].

To explain the long-lasting dynamics of slow earthquakes, numerous observations suggest the crucial role of viscous fluid around faults[11–13]. Some numerical studies propose the effects of the diffusional dynamics of fault size evolution[14] and heterogeneous stress distribution[15]. Furthermore, in the past decade, many gouge friction experiments have successfully reproduced unstable sliding with slow slip and/or rupture velocities[16–23].

However, the multiple unique statistical laws of slow earthquakes and their origin have scarcely been investigated through experiments. This is due to two main difficulties: reproducing the observed statistics within finite system sizes, and directly observing materials' internal microstructures during gouge friction experiments. Despite such difficulties, a stick-slip experiment during antigorite serpentinite gouge dehydration successfully demonstrates the roughly linear scaling between stress drop and duration, corresponding to $M_0 \propto T$ (ref. 17). Owing to the lack of experimental data simultaneously reproducing the multiple statistical properties, the mechanism responsible for the slow earthquake statistics remains controversial. Experimental elucidation of their enigmatic origin would clarify the relationship between slow and regular large earthquakes[2,24], which control the energy balance of plate tectonics[13].

To relate the macroscopic statistics of slow earthquakes to the microscopic dynamics of fault gouge[23,25–28], a combined approach of stick-slip experiments[29] and in situ fabric observations using soft matter has several advantages. Fault gouge materials generating slow earthquakes are presumed to be rich in low-modulus hydrous clay minerals[30], high-pressure pore fluids[12], and/or viscous phases associated with brittle-ductile transition zones[31]. These factors make a fault low-frictional and soft. Therefore, the mixture of low-friction soft granular material and viscous liquid demonstrates behavior analogous to that of gouge[32–34]. Experiments with soft materials at room temperature and atmospheric pressure allow us to arrest ruptures within

Department of Earth and Space Science, The University of Osaka, 1-1, Machikaneyama, Toyonaka 5600043 Osaka, Japan. ✉e-mail: sasaki.geoscience@gmail.com; katsuragi@ess.sci.osaka-u.ac.jp

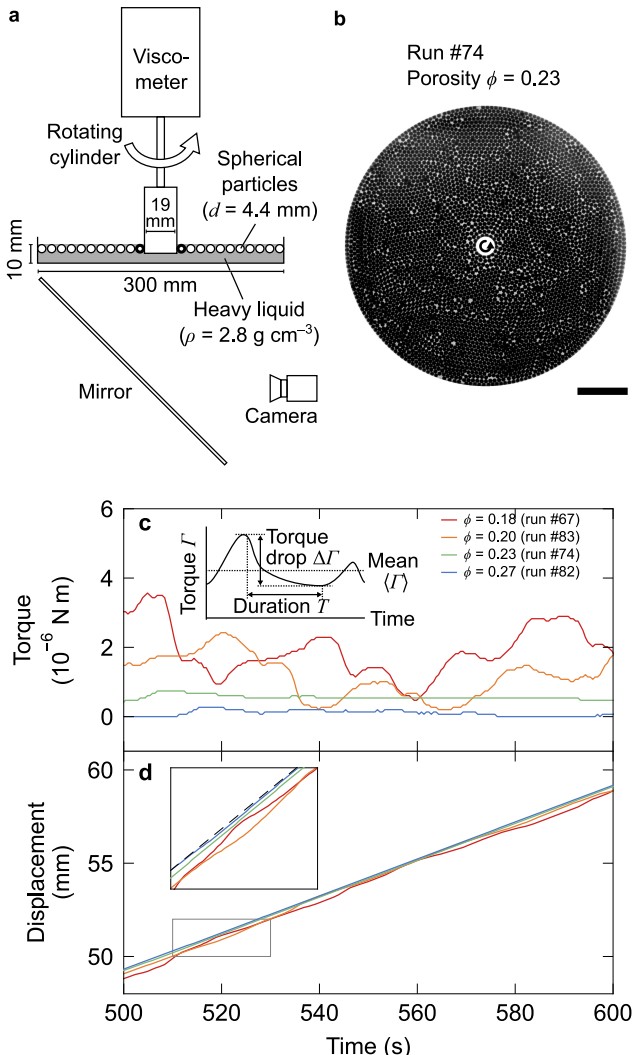

**Fig. 1 | Experimental setup and observed stick-slip behavior. a** Quasi-two-dimensional granular shear apparatus. Spherical hydrogel particles (4.4 mm in diameter) are floated on a heavy liquid (2.8 g cm$^{-3}$). A cylinder, coupled to a viscometer motor through a torsion spring, rotates at a constant angular velocity of 0.60 deg s$^{-1}$. Sixteen particles (indicated by thick circles) are fixed to the cylinder surface. Particle motions were tracked in situ from the bottom of the container. The floating particle layer (300 mm in diameter) is penetrated by the rotating cylinder ($2R = 19$ mm in diameter) in a 10 mm thick liquid layer. The schematic of the figure is not drawn to scale. **b** Representative image at a porosity $\phi$ of 0.23. The circular arrow indicates the direction of cylinder rotation. Note that tens of out-of-plane particles remained in the system, which were unremovable. Scale bar: 50 mm. **c** Temporal evolution of torque for different porosities ($\phi = 0.27$–0.18). Inset shows the definitions of torque drop amplitude ($\Delta\Gamma$) and duration ($T$). **d** Temporal evolution of tangential displacement at the cylinder surface, with color coding corresponding to panel (**c**). Inset shows an enlarged view of the boxed area. The dashed line represents the steady shear rate of 0.099 mm s$^{-1}$.

the finite system size[21,35] and track each particle. In such a granular system, porosity not only represents the ratio of fluid or ductile phase, but also controls pressure, fault strength, fluid migration, and slip statistics. Accordingly, this serves as a useful framework for understanding the mechanism of slow earthquake faulting.

In this work, we experimentally investigated the statistics of low-friction soft granular slip events and the effect of porosity on them. We conducted quasi-two-dimensional (2D) rotary shear experiments on a monodisperse hydrogel granular layer with low friction and low elastic modulus, which floats on a viscous lubricating liquid surface as shown in Fig. 1a (see Methods for details). The experimental system with a

constant volume enables the systematic variation of the bulk porosity $\phi$ and direct tracking of particle motion (Fig. 1b), both of which intrinsically govern the behavior of fluid-retaining faults. We captured hundreds of stick- and slow-slip events under shear, which become more pronounced with decreasing porosity (Fig. 1c, d). Slip events in this low-friction soft granular system simultaneously follow two statistical characteristics of slow earthquakes: the moment-duration scaling and size distribution[2,3], as shown in Fig. 2. Based on the in situ particle observation, shear localization at lower porosity (Fig. 3) explains these porosity-dependent statistics of slip moment (Fig. 4) as well as the depth distribution of earthquakes across the spectrum of their types (Fig. 5).

## Results

### Moment statistics of slip events
The fluid-lubricated soft granular layer under shear exhibits torque fluctuations corresponding to stick-slip events (Fig. 1c, Supplementary Fig. 1). In this lubricated system, the stick-slip behavior involves both partially decoupled stable shear during the stick phase and unstable slip with torque drops, accommodating total displacement similar to that in slow earthquakes (Fig. 1d). This transitional behavior between creep and stick-slip is typically observed in lubricated granular systems[32,33]. At a porosity of $\phi \gtrsim 0.24$, creep motion virtually dominates all deformation, whereas at $\phi = 0.18$, the unstable slip events account for 32% of the total displacement. A decrease in porosity to approximately 0.20 causes a transition from creep to stick-slip behavior (Fig. 1c, d). The slip rate exceeds the steady shear rate, shown by the dashed black line in the inset of Fig. 1d. The frequency of slip events roughly increases with the porosity decrease (Supplementary Fig. 2, Supplementary Table 1). The decrease in porosity also causes an increase in both the mean torque strength and its drop amplitude (Fig. 1c, Supplementary Fig. 3a, b), which are defined in the inset of Fig. 1c. The maximum torque at the minimum porosity remains sufficiently low, resulting in no particle failure. The porosity dependence of the torque drop amplitude remains consistent across system sizes ranging from 100 to 300 mm in diameter (Supplementary Fig. 3b).

We statistically analyzed the moment $M_O$ and its rate $\dot{M}_O$ of the slip events, calculated from the torque drop amplitude, drop duration, and observed slip area (see Methods). Consistent with the variation in the torque drop amplitude (Supplementary Fig. 4a, b), a decrease in porosity from 0.41 to 0.18 progressively increases both the amount and rate of moment release (Fig. 2).

Statistically, the moment $M_O$ follows an exponential size distribution, regardless of porosity (Fig. 2a). This is in contrast with regular earthquakes[6,7] and dry, confined granular materials[36–38], following a power-law size distribution, i.e., a scale-invariant process. Rather, our result is consistent with slow earthquakes following an exponential size distribution or a power-law size distribution with a higher exponent, both of which are difficult to statistically distinguish[3,10]. In addition, a power-law size distribution truncated with an exponential function seems to be exhibited at a slower shear rate (Supplementary Fig. 5a). The characteristic size (slope) of the exponential size distribution decreases with increasing porosity $\phi$, consistent with previous experimental studies[39]. These exponential size distributions represent a scale-limited process, characterized by the area or amount of slip in slow earthquakes[3].

Our experiments also revealed a nearly linear moment-duration scaling of $M_O \propto T$, indicating a constant moment rate $\dot{M}_O$ (Fig. 2b). A slower shear rate results in approaching a linear moment-duration scaling (Supplementary Fig. 5b). An increase in timescale (slow shearing) consistently induces the linear scaling between size and duration in other granular experiments[37] and seismic observations[2], although the underlying mechanism remains unclear. Crucially, our laboratory experiments simultaneously reproduce the exponential size

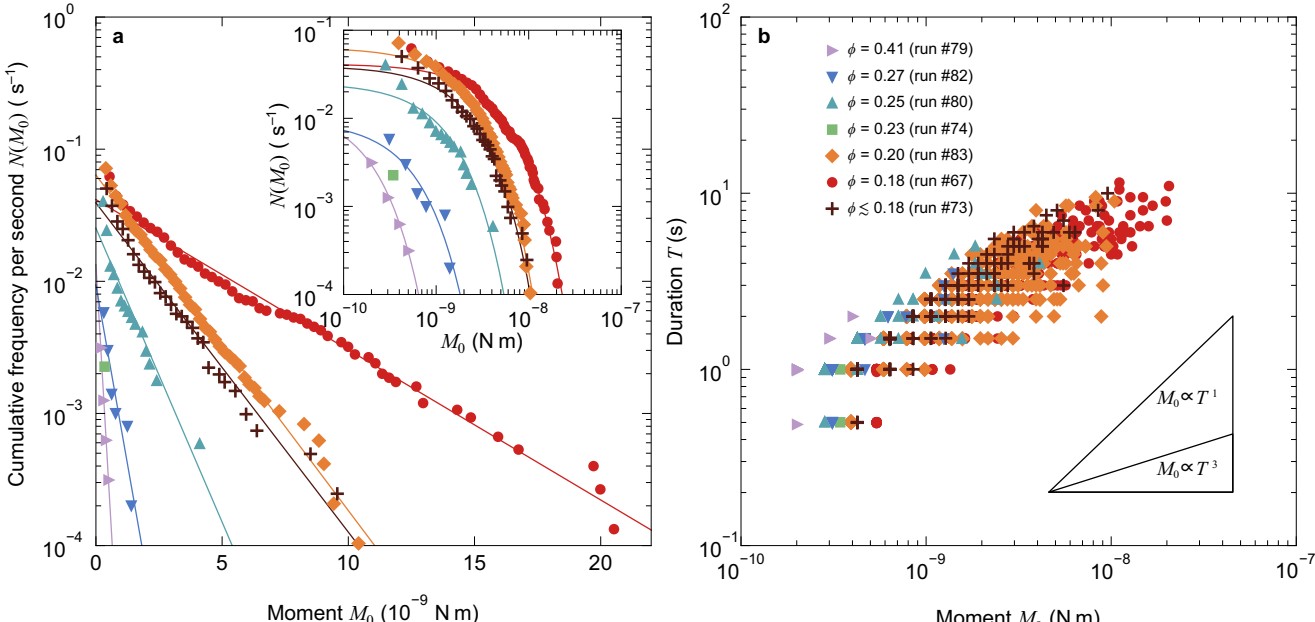

**Fig. 2 | Statistical properties of the released moment. a** Cumulative frequency distribution of the moment (per second). The linear trends on a semi-log scale demonstrate exponential distributions, with fitting curves shown as solid lines. The inset shows the same data in a log-log plot. The colors and symbols correspond to those in (**b**). **b** Relationship between moment and duration on a log-log scale for porosities ranging from $\phi = 0.41$ to $0.18$.

distribution and linear moment-duration scaling, key features of slow earthquakes.

## In situ granular motions

To investigate the representative particle movements contributing to the observed moment statistics, we captured in situ particle arrangements as binary images every 0.1 s (Fig. 1b). Fig. 3a–c shows the comparison of the time-lapse images for $\phi = 0.23$, 0.20, and 0.18, mapping the standard deviation of temporally varying binary values at each pixel over approximately 1100 s and 660 degrees of rotation (see Methods). These images reveal three types of regions formed at any porosity: the central shear band with active motion (continuous black region), the surrounding vibration region with particle drift shorter than the particle diameter, and the outer fixed region with no noticeable motion. The particles principally slip on the boundary plane within the granular layer between the shear band and the surrounding vibration region. When dry, angular quartz sand grains are sheared in a setup similar to ours, they also exhibit the central shear band and intermittent radial slip propagation into the surrounding region[27]. This implies that the granular fabric we observed is qualitatively consistent, regardless of lubrication and particle shape. Following the structure of granular fault[40], the outer fixed region and the others correspond to the damage zone and the fault core, respectively; the shear band corresponds to the principal slip zone of a natural fault[41].

The shear band becomes more localized at lower porosities (Fig. 3a–c). We obtained a linear relationship between the shear band thickness $w$ and $\phi$ (Fig. 4a). As $\phi$ decreased from 0.41 to 0.18, $w$ decreased from $\gtrsim 9d$ to $2d$ with the particle diameter $d$. The shear band thickness of less than ten particle diameters, an approximate typical value for dry frictional conditions[27,42], is associated with the shorter correlation length of strain under low elastic modulus[43] and low friction. This shear localization is also observed in lubricated granular particles with an increase of pore-fluid viscosity[32,33]. Notably, the released moment exhibits larger values under conditions of lower porosity despite the shear localization (Fig. 4b). Meanwhile, the porosity within the localized shear band remained constant at 0.3, which is higher (richer in fluid content) than the bulk $\phi$, except under the

highest $\phi$ value in run #79 (Supplementary Fig. 3e). Note that we only varied the bulk porosity $\phi$; its spatial distribution spontaneously evolves and stabilizes under steady state.

Additionally, the surrounding vibration region exhibits radial anisotropy that persists throughout the entire run time (Fig. 3a–c). This implies the presence of fixed force-chain structures[44], which supports the shear band without their significant rearrangement during slip events. Furthermore, the outer fixed region implies that the mechanical correlation length is sufficiently short relative to the entire system size. This ensures that slip and rupture propagation are confined within the shear band, not extending throughout the entire system. The outer fixed region can apply low normal stress (radial pressure) to the shear band that has higher porosity, constraining its thickness $w(\phi)$.

We further examined the distribution and evolution of slip planes associated with each event. Figure 3d illustrates the typical types of particle motion with instantaneous displacement for different events at $\phi = 0.20$. The black areas indicate particle differential motion over 2 s. Quantifying the number of these moving particles during each event reveals that slip events predominantly occurred on slip planes of similar size within the shear band for each porosity, independent of the moment $M_0$ (see Methods; Supplementary Fig. 6). This justifies defining a representative constant value of slip plane area $S(w(\phi))$ for each porosity. Figure 3e shows the temporal evolution of slip motions over 17.2 s, corresponding to the torque drops. The blue arrows indicate the rupture front, which is identified as the boundary between areas of particle mobility and the quiescent surroundings. Its propagation timescale is approximately 1 s. This rupture time is relatively shorter than the overall slip event duration $T$ in this lubricated system. Thus, the low-friction and soft granular motion is primarily characterized by shear localization with decreasing porosity and slip on a plane of constant area for each porosity, regardless of the moment. These unique characteristics of the granular dynamics can explain the moment statistics.

## Possible mechanisms for the statistics

The low-friction soft granular system exhibits both the exponential size distribution and linear moment-duration scaling (Fig. 2). These

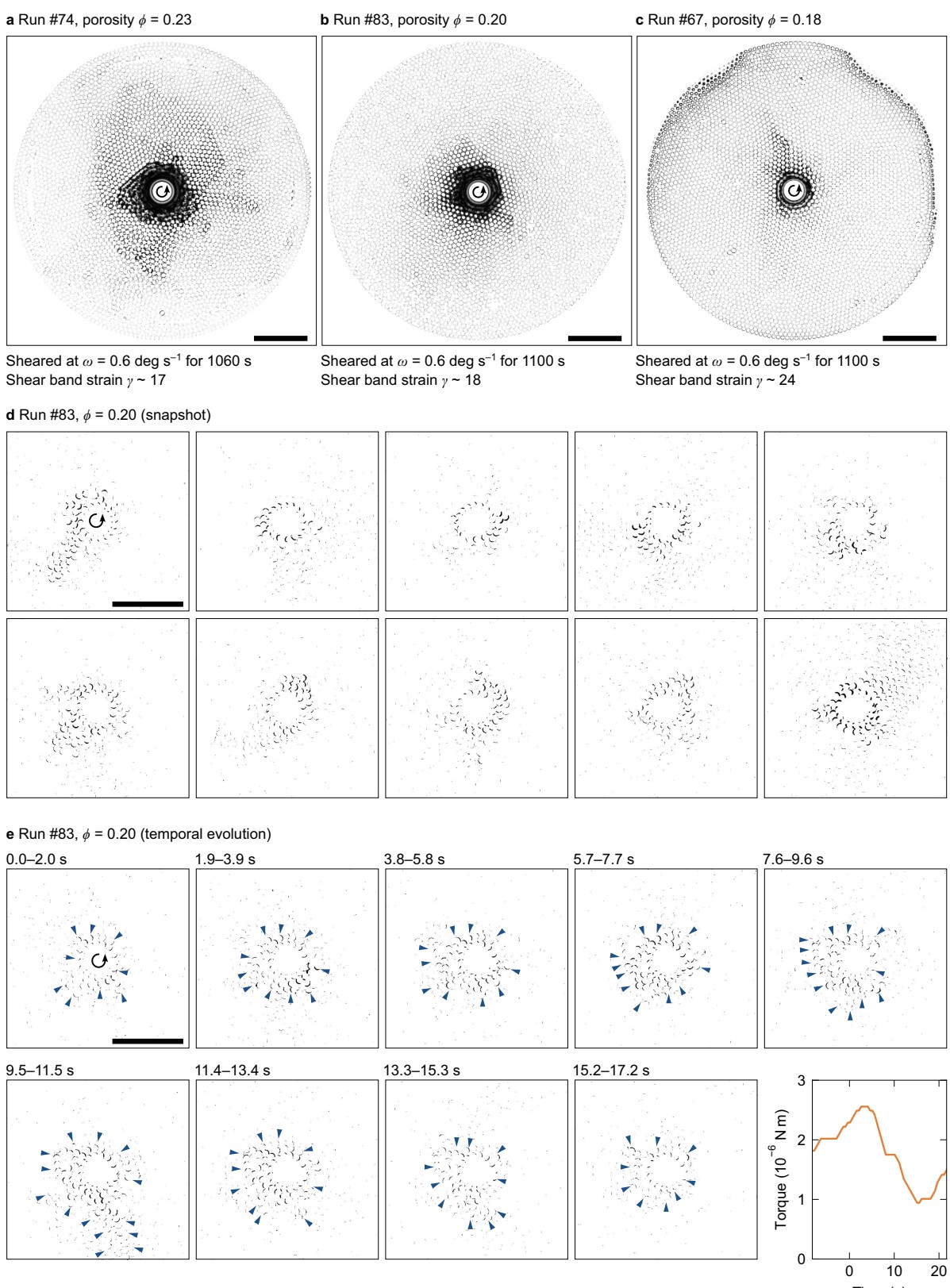

**a** Run #74, porosity $\phi$ = 0.23

Sheared at $\omega$ = 0.6 deg s$^{-1}$ for 1060 s
Shear band strain $\gamma$ ~ 17

**b** Run #83, porosity $\phi$ = 0.20

Sheared at $\omega$ = 0.6 deg s$^{-1}$ for 1100 s
Shear band strain $\gamma$ ~ 18

**c** Run #67, porosity $\phi$ = 0.18

Sheared at $\omega$ = 0.6 deg s$^{-1}$ for 1100 s
Shear band strain $\gamma$ ~ 24

**d** Run #83, $\phi$ = 0.20 (snapshot)

**e** Run #83, $\phi$ = 0.20 (temporal evolution)

0.0–2.0 s   1.9–3.9 s   3.8–5.8 s   5.7–7.7 s   7.6–9.6 s

9.5–11.5 s   11.4–13.4 s   13.3–15.3 s   15.2–17.2 s

**Fig. 3 | Shear band structures and particle motions responsible for torque drop events. a**–**c** Shear band structures. Darkness corresponds to the standard deviations of binary videos recorded over approximately 1100 s at porosities $\phi$ = 0.23, 0.20, and 0.18, respectively. Nominal shear strains $\gamma$ in polar coordinates, estimated from shear band thickness $w$, are approximately 17, 18, and 24, respectively. Note that **c** shows increased noise at edges owing to lighting conditions. Scale bars: 50 mm. **d** Typical patterns and area size of intermittent particle motions during different events. Black areas represent particle-sweeping areas, obtained from binary image differences at a 2-second interval. A scale bar of 50 mm is common to all images. **e** Time series of particle motions corresponding to the consecutive torque drops. Image processing and the scale bar are the same as in (**d**). The time 0 s corresponds to a run time of 2228.3 s in run #83. Blue arrows indicate the rupture front. Torque data corresponding to the images are shown in the last panel. An animated version is available online as Supplementary Movie 1.

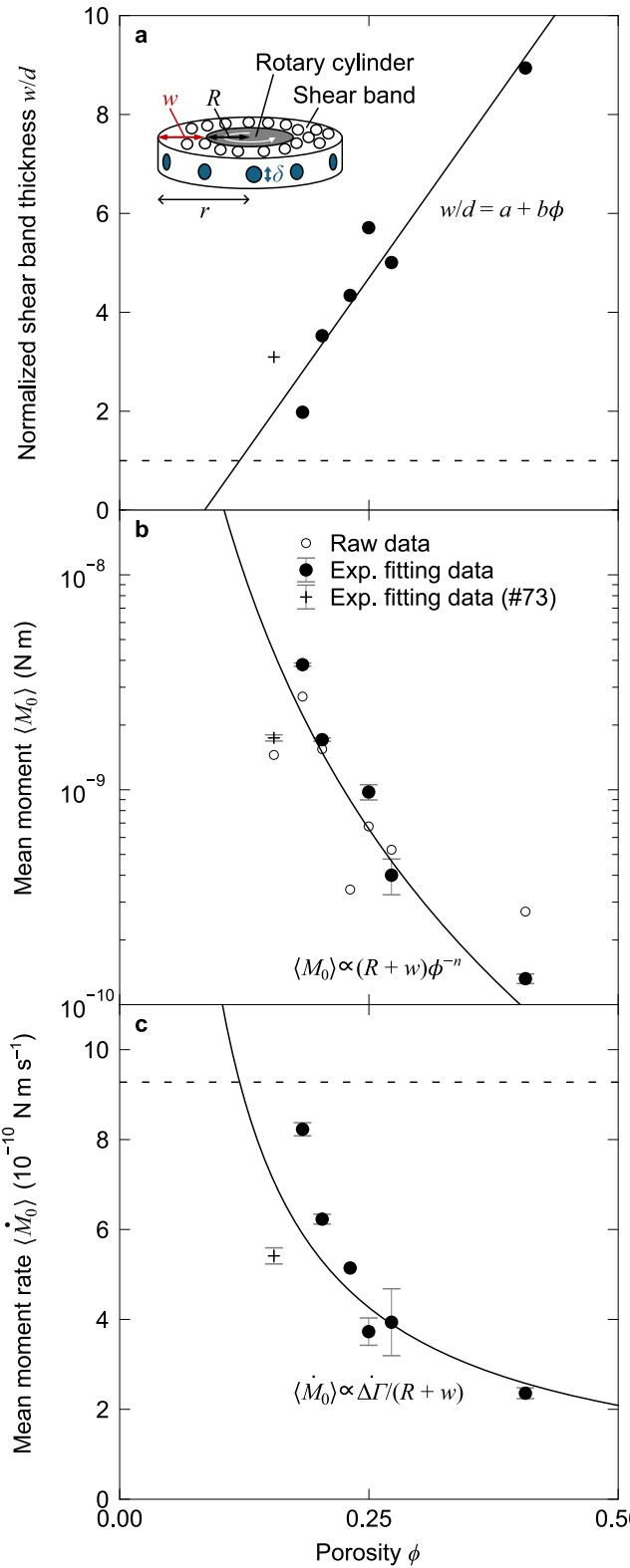

**Fig. 4 | Effects of shear localization on the moment. a** Shear band thickness normalized by the particle diameter as a function of porosity. The data points represent experimental data. The solid line represents the linear fit with $a = -10.7$ and $b = 125$, excluding the cross symbol at the lowest porosity in run #73 ($0.15 \lesssim \phi \lesssim 0.18$), where many particles were not confined to the same monolayer. This exceptional datum is provisionally plotted at $\phi = 0.15$. The dashed line indicates the minimum thickness, $w/d = 1$. The inset indicates the schematic model of the shear band with its outer boundary plane at a radial distance, $r = R + w$, where each contact patch has a diameter $\delta$. **b** Mean moment as a function of porosity. The black and white circles represent the mean values from exponential fitting and raw data, respectively. The solid line represents the fitting curve of all the solid black circles with $n = 5.0$ (Equations (1) and (8)), except for the cross symbol as in (**a**). Error bars represent standard error based on 4–49 different moment sizes for the exponential fitting, also for (**c**). **c** Mean moment rate as a function of porosity. The black circles represent the mean values and the solid line indicates the reference curve of $\dot{M}_0 = M_0/T$ with constant slip velocity (Equations (1) and (2)). The dashed line indicates the maximum value at $w/d = 1$.

collapse[45]. The stress of both force chains and consisting particles follows an exponential distribution[44,46]. In contrast to our system, frictional particles under dry conditions and higher normal stress follow power-law size distributions with exponential truncation, regardless of system dimension[36–38]. In the frictional systems under normal stress, the force chains form networks that can cascade into critical slip avalanches. However, lubrication and/or low normal stress (radial pressure) in this study inhibit the formation of force chain network, thereby restricting slip size to the scale of individual particles and suppressing moment release during slip events (Fig. 5a–c). When we substituted hydrogel particles with more rigid glass beads with frictional surfaces, we observed a superposition of exponential and power-law size distributions under lubrication (Supplementary Fig. 10a). This distribution is also observed under a lower shear rate condition (Supplementary Fig. 5a), which implies the distinction of individual events owing to complete relaxation and quasi-static loading. These results demonstrate that low friction, as well as low elastic modulus, low normal stress, and slower relaxation than shear rate, plays a crucial role in limiting the mechanically correlated length and preventing cascade collapse events. Consequently, granular systems with these properties manifest the exponential size distribution that reflect the heterogeneous stress distribution of individual force chains (Fig. 5a, b).

The characteristic scale of the size distribution can be determined by the balance between the enhanced force chain strength and localized slip area with normal stress. To explain the relationship between the mean moment and porosity, we assume that slips occur repeatedly on the outer boundary plane of the cylindrical shear band, as shown in the inset of Fig. 4a. Although each particle slips within the buckling force chains, we consider that most of the strain energy is dissipated by the slips on the shear-band boundary plane, resulting from collective particle rearrangements within the force chains. The shear band has a thickness $w$ and a diameter $\delta$ of the interparticle contact patch. The seismic moment is defined as $M_0 = \mu S D$, where $\mu$ is the shear modulus, $S$ is the slip area, and $D$ is the slip displacement[47]. This moment resulting from the torque drop with an amplitude $\Delta\Gamma$ and associated slip with a plane area $S = 2\pi(R + w)\delta$ can be calculated as,

$$M_0 = \frac{\delta}{C(R+w)}\Delta\Gamma = \frac{2\pi\delta^2}{C}(R+w)\Delta\sigma, \qquad (1)$$

with the shear stress drop $\Delta\sigma$, a geometrical constant $C$, and the rotating cylinder radius $R$ (see Methods). In this equation, the mechanical force drop $\Delta\Gamma/(R + w)$ is converted into the moment on the slip plane with the characteristic width $\delta$. Equation (1) adequately explains the porosity-dependent characteristic moment $\langle M_0 \rangle$, with the

two statistical laws remain consistent across various experimental conditions (Supplementary Table 1), including polydispersity (Supplementary Fig. 7), liquid viscosity (Supplementary Fig. 8), and system size (Supplementary Fig. 9). This universality ensures that the two statistical laws can be explained by a simple model, as follows.

First, the exponential size distribution (Fig. 2a) should emerge from the strength of the granular layer, which is generally supported by force chains[44] and induced to slip by their buckling

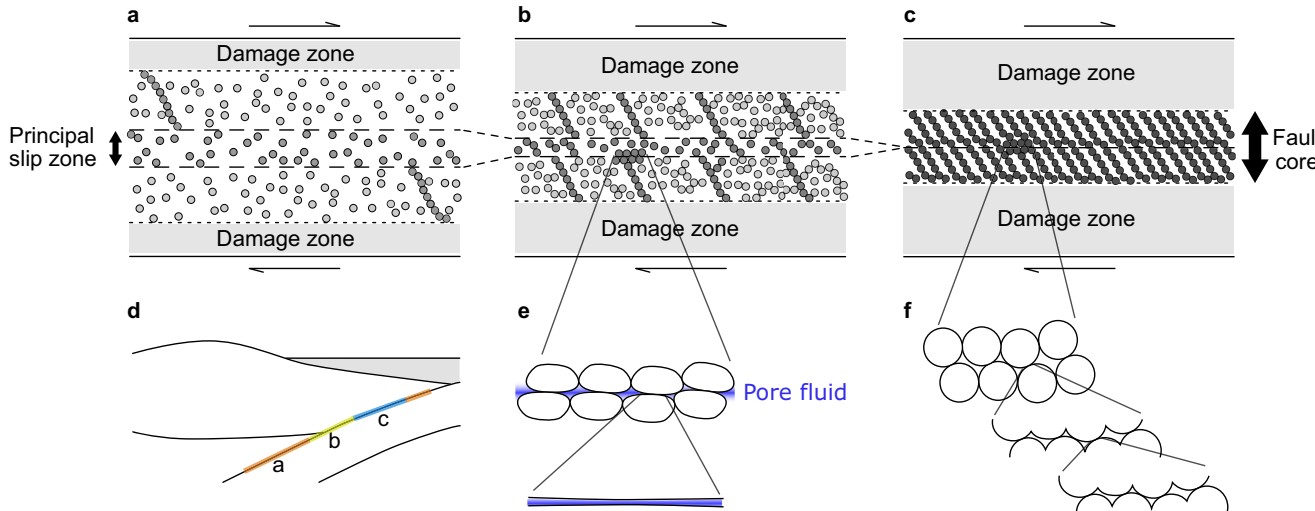

**Fig. 5 | Schematic model of the granular fault core structures in slow and regular earthquakes.** Granular-scale models of slow (**a**, **b**) and regular earthquakes (**c**), along with an alternative interpretation based on microscopic patch structures (**e**, **f**). **a** Deeper region on the downdip transition zone (short-term slow earthquakes). A broad principal slip zone (shear band) is supported by long, weak force chains that are sparsely distributed across the fault core. The gray particles represent brittle grains/blocks, with the darkness indicating accumulated stress. The white areas represent pore fluid and/or ductile matrix (same for the following). **b** Shallower region on the downdip transition zone (long-term slow earthquakes).

A localized principal slip zone is supported by short, strong, and densely distributed force chains. **c** Seismogenic zone. Frictional, rigid force chains form percolating networks throughout the fault core, where growing ruptures lead to the generation of critical avalanches. **d** Cross-section of a subduction zone and plate interface. **e** Lubricated soft granular contact state in the fault core of slow earthquakes as (**b**). Pore fluid and/or ductile matrix concentrates within the principal slip zone, and soft particles make contact patches smoother and flatter. **f** Frictional rigid granular contact state in the fault core of regular earthquakes as (**c**). Rigid particles with rough surfaces exhibit fractal patches.

fitted scaling $\langle \Delta\sigma \rangle \propto \phi^{-n}$ ($n = 5.0$) and the fitted linear relationship $w(\phi)$, assuming constant $\delta$ (solid line in Fig. 4b). $\langle\rangle$ denotes the mean value. This indicates that despite shear localization, the increases in normal stress and resultant shear stress predominantly contribute to the moment. This is consistent with the interpretation that the force chain strength governs the moment statistics[40]. Granular materials typically exhibit a rigidity phase transition and normal stress enhancement with decreasing porosity[48], which is also observed in this study (Supplementary Fig. 3c). Thus, the size distribution is governed by the exponentially distributed stress drop $\Delta\sigma$ and its corresponding slip displacement $D$, which vary with porosity (see Methods; Supplementary Fig. 4b).

Second, the linear moment-duration scaling, $M_0 \propto T$ (Fig. 2b), indicates a violation of self-similarity (fractal) holding in regular earthquakes, where $M_0 \propto T^3$ (refs. 4,5). In regular earthquakes, self-similarity emerges from the relationship between slip area $S$ and displacement $D$, where larger slip areas require greater displacements as $D \propto \sqrt{S}$. In contrast, our low-friction soft granular system exhibits the opposite trend, where $D$ increases as $S(w(\phi))$ and $\phi$ decrease (Supplementary Fig. 4c). We observed that the temporally-averaged slip velocity of the cylinder $\overline{V}$ ($\propto D/T$) remains constant regardless of porosity (Supplementary Figs. 3d, 11), resulting in the linear moment-duration scaling $M_0 \propto \mu S \overline{V} T \propto T$ with constant $S(w(\phi))$ for each porosity (Fig. 4a). Furthermore, when we investigated the effect of elastic modulus on the moment-duration scaling using glass beads, we observed a nonlinear scaling of $M_0 \propto T^2$, especially in the high stress regime (Supplementary Fig. 10b). This suggests that the particle modulus or stress level is related to the constancy of slip area and velocity, which realizes the linear moment-duration scaling. Rigid particles, such as glass beads and fault gouge grains in the seismogenic zone, can preserve the fractal structures of force-chain networks at the percolation threshold[49] (Fig. 5c) or of rough interparticle patches (Fig. 5f), resulting in the cubic moment-duration scaling $M_0 \propto T^3$. In contrast, low-modulus particles and/or low normal stress cause the monotonous structures of force chains (Fig. 5a, b) or interparticle patches (Fig. 5e), violating the self-similar scaling.

Note that, however, the effects of lubrication and system dimension cannot be ruled out. In particular, the self-similar moment-duration scaling $M_0 \propto T^3$ is not realized in a 2D system due to anisotropic rupture propagation. The fault plane with a finite width, corresponding to the width $\delta$ in our granular layer, exhibits a strong dependence of the transition in the moment-duration scaling on stress heterogeneity[15]. Any of these cases potentially leads to a loss of fractality in the distributions of displacement and patch area of slip by the force chain collapse (Supplementary Fig. 4c, 6), resulting in the linear moment-duration scaling $M_0 \propto T$ (Fig. 2b).

The constant moment rate $\dot{M}_0$ ($= M_0/T$) in the linear moment-duration scaling can be expressed in terms of the system's torsional stiffness $K$ (N m deg$^{-1}$) using,

$$T = \frac{D}{\overline{V}} = \frac{c\Delta\Gamma/360K\overline{V}}{1 - c\omega/360\overline{V}}, \tag{2}$$

with the tangential slip displacement $D = \overline{V}T$, circumference $c = 2\pi R$, and angular velocity $\omega$ (deg s$^{-1}$) of the cylinder (see Methods). The numerator of this equation represents the elastic shear deformation of the system, whereas the denominator applies a correction for the steady rotation of the motor during each slip. Using Equations (1) and (2), this model effectively explains the porosity-dependent mean moment rate $\langle \dot{M}_0 \rangle$, with the fitted shear band thickness $w(\phi)$ and the constant slip velocity $\overline{V}$, assuming constant $\delta$ (solid line in Fig. 4c). $\overline{V}$ in Equation (2) is constant in this study (Supplementary Figs. 3d, 11), consistent with the constant torque drop rate $\Delta\Gamma/T$ and exponentially distributed slip displacement $D$ (Supplementary Fig. 4b, c). As the shear band thickness has a minimum value of $w \sim d$ in this study, the moment rate should approach a maximum limit (dashed line in Fig. 4c). In natural fault systems, the principal slip zone thickness for each event typically ranges from 1 mm to 1 cm in the fault core[41]. This characteristic length scale of shear localization possibly constrains both the moment scale and also the maximum moment rate observed in slow earthquakes[2]. Our quantitative models for both the exponential size distribution and the linear moment-duration scaling provide a self-

consistent framework for slow earthquake statistics, interpreted as the suppression of the force-chain network percolation and fractal patch formation.

## Discussion

The origin of slow earthquake statistics can be attributed to the heterogeneous stress distribution and monotonous patch structure at the scale of brittle grains or blocks within the fault zone. This emerges from fluid-lubricated, soft hydrous clay gouge within the finite-width fault zone. The following discussion provides the supporting evidence.

Our results of slow earthquake statistics indicate that lubricated soft particles under low radial pressure and with finite fault width exhibit local force-chain collapse and monotonous structures of interparticle patches. In this context, particles comprising force chains can be considered as gouge grains or even as brittle blocks across multiple scales[50,51]. Low radial pressure and finite fault width correspond to low effective normal stress due to pore fluid pressure and depth limitation of the seismogenic zone, respectively. These factors mitigate the cascading processes in the on- and off-fault directions, as well as the self-similarity of faulting, both of which can be applied to fault gouge materials.

In gouge friction experiments for slow slip phenomena, the linear scaling of stress drop and resultant moment, roughly proportional to duration, is obtained with dehydrating antigorite serpentinite[17]. This result is similar to ours (Supplementary Fig. 4b). The same study also reports a constant slip velocity, smaller stress drop, and longer duration under dehydration, which are observed in both our system and another viscously lubricated granular shear[32]. These consistently correspond to slow earthquake characteristics and the linear moment-duration scaling of $M_0 \propto T$ in the presence of fluid. In contrast, most gouge friction experiments under conditions ranging from dry to 100% relative humidity and high normal stress show the negative correlation between stress drop and duration across slow to fast slip events[19,21,23]. The saturation state of the liquid volume confined within pores, as well as normal stress, might play a role in this difference.

We could not find any previous work on the size distribution in slow slip gouge friction experiments, as most studies have focused on slip velocity. However, acoustic emissions during glass bead stick-slips exhibit a power-law size distribution with a slightly higher exponent, known as the *b*-value[7], for lower velocity slip events under lower stress[52]. This is consistent with the stress dependency of the *b*-value in rock sample acoustic emissions[53]. Moreover, the power-law size distribution of rock samples demonstrates a lower *b*-value on a localized fault plane with smaller stress heterogeneity acting upon it[54] (Fig. 5c). Therefore, slow earthquakes in the laboratory, occurring under conditions of low normal stress and stress heterogeneity at the scale of granular force chains, are anticipated to exhibit a higher *b*-value, leading to an exponential-like function observed in our experiments.

Moreover, a critical role of force chain dynamics and its dilatancy/compaction effect has also been recognized by gouge friction experiments[23,25,26,28]. The higher porosity within our localized shear band also implies the dilatancy effect. These studies support our interpretation that force chain development and buckling collapse[45] correspond to the loading and stress drop phases, respectively.

The mechanism of slow earthquakes proposed by gouge friction experiments is based on frictional instability, which appears to differ from our scenario despite many consistencies. However, the low normal stress condition in our experimental system is consistent with the slow-slip gouge friction experiments characterized by high normalized system stiffness owing to low critical stiffness[16–18,20,21]. When the explanation that transitional slip near the boundary between stable and unstable frictional sliding corresponds to slow slip phenomena[18,20,55] is accepted, the stiffness condition in this study possibly lies near the stability boundary. As the critical stiffness reflects the characteristics of rate- and state-dependent friction[56,57], the

persistence of slow slip velocities has been explained within this framework[16,17,55,58]. Accordingly, the bulk frictional properties and relevant porosity evolution of our lubricated soft granular system warrant further investigations in this context.

Applying our granular model to fault gouge, the porosity dependence of moment statistics can qualitatively explain the depth distribution of both slow and regular earthquakes along the subducting plate interface[59]. Figure 5a–c schematically illustrate the fault cores in the deeper region on the downdip transition zone, in the shallower region on the same zone, and in the seismogenic zone, respectively (Fig. 5d). Our results suggest that with much fluid or ductile matrix at higher porosity, shear localizes within the principal slip zone[33], which is mechanically supported by force chains comprising the rest of the fault core (Fig. 5a). The individual force chain locally collapses without cascading up, corresponding to slow earthquake events. Meanwhile, the reduction in fluid-filled porosity and localization of the principal slip zone facilitate the formation of percolating force-chain networks over the fault core, which critically induce the macroscopic slip avalanche behavior, i.e., regular earthquakes (Fig. 5c). In this sense, the seismogenic zone has the lowest porosity, with the fewest pores and ductile phases among all depths. As shown in Fig. 5b, the distribution of more long-term slow slip events in the shallower region on the transition zone, adjacent to the seismogenic zone[59], might be explained by lower porosity (ductile phase ratio) with longer and slightly faster slip than in the deeper region (Fig. 2b, Supplementary Fig. 11).

Furthermore, the dependence of moment on porosity should be observed when local porosity changes according to the pressure of migrating pore fluid. Tectonic tremor events, a type of slow earthquake, exhibit a temporal change in the mean size and occurrence rate, both of which tend to be positively correlated[60]. This correlation is consistent with our results (Figs. 2a, 4b, Supplementary Fig. 2), while the seismological study claims that the size and rate decrease with pore fluid pressure reduction.

Building upon our findings, this simple experimental system can facilitate further promising investigations into the effects of shear rate, pressure in 3D, system geometry[40,61], granular fabric[23,28], and rheology[34,62]. As an example of unresolved inconsistency, a simple shear setup with evolving aspect ratio also exhibits exponential size distributions, while its porosity dependence is opposed to ours[61]. Our study lays the groundwork for addressing the scaling between stress drop and slip velocity[19,21,22], the fracture energy scaling of slow earthquakes[63,64], and the modeling of seismic moment as well as shear band thickness[51] through nondimensionalization. Based on our findings, the discovery of aligned brittle grains or blocks forming force-chain structures in geological outcrops is anticipated in slow earthquake fault zones[50,51]. Spatiotemporal observations of slow earthquake statistics could help monitor fluid-filled fault zones[60], fault strength or stress[24], and shear localization that could lead to devastating regular earthquakes such as the 2024 $M_{\rm w}$ 7.5 Noto earthquake[65].

## Methods

We conducted rotary shear experiments on a floating granular monolayer, showing stick-slip behaviors. We measured torque and recorded the particle arrangement in situ under various porosities and other conditions (Supplementary Table 1). The temporal evolutions of mechanical torque and visual data were statistically quantified.

### Experimental setup

The experimental setup is shown in Fig. 1a. We sheared a layer of particles floating on a heavy liquid surface in a cylindrical container (radius $R_c$) at room temperature and atmospheric pressure by inserting a rotating cylinder (radius $R$) into the center of the layer. The floating particles comprise an approximately one-particle-thick monolayer, enabling the tracking of all particles with a camera. This

quasi-two-dimensional granular system removes the effects of basal and interparticle frictions, leading to enhanced particle dispersion. This lubricated granular layer is appropriate to simulate low effective normal stress owing to high pore pressure, widely suggested as a cause of regular and slow earthquakes[12,22]. We used a cylindrical Couette system in two configurations, one with $2R_c$ = 300 mm and $2R$ = 18.84 mm, and the other with $2R_c$ = 100 mm and $2R$ = 18.72 mm. The ratio of the channel gap ($R_c - R$) to the particle diameter ($d$ = 4.4 mm) was 32 and 21, respectively (Fig. 1b). The same 16 particles used as the experimental sample were glued in a single row around the side surface of the cylinder to roughen it, except for runs #27–31 with 17 particles. In the case of a bidisperse mixture described below (run #77), hydrogel particles ($d$ = 4.4 mm) were used for this purpose.

We used three types of particles as a fault gouge analogue: opaque spherical polymer hydrogel particles (Leaf Corporation, Bio Beads, PEG) and spherical glass beads of two sizes. Hydrogel particles have a diameter of $d$ = 4.4 mm with a standard deviation of 0.1 mm, with lower friction (coefficient of < $10^{-1}$) and lower modulus (~ kPa)[66]. The two types of glass beads have $d$ = 4.1 or 2 mm, respectively, with higher friction (coefficient of $\gtrsim$ 0.2)[67] and higher modulus (~ GPa). To align particle centroids within the same plane on the liquid, we used a single type of monodisperse particle in each experimental run. To investigate the effect of particle arrangement, a bidisperse mixture of the hydrogels and 2 mm glass beads was also used. For preparation, we immersed the hydrogel particles in the heavy liquid solution for more than one week, as the hydrogel particles change volume owing to osmotic pressure depending on the concentration of the solution. The osmotic effect made the particles shrink by 2–3% in diameter 5 days after starting the permeation of the water-immersed particles.

To float the particles, we used two types of liquid. A transparent solution of sodium polytungstate ($Na_6(H_2W_{12}O_6)$, TC-Tungsten Compounds GmbH, purity of > 99%) was prepared with a density and a viscosity of approximately 2.8 g cm$^{-3}$ and $2 \times 10^{-2}$ Pa s, or slightly above, respectively. Exceptionally, in runs #56 and #57, the density was approximately 2.6 g cm$^{-3}$ and the relevant viscosity was slightly lower. To investigate the effect of viscosity, another solution was also prepared by adding a thickener of carboxymethyl cellulose (As One Co., Ltd., CMF-150) with a density of 2.8 g cm$^{-3}$ and a viscosity of 1 Pa s. The densities made the particles remain suspended just below the liquid surface level, as schematically shown in Fig. 1a. To prepare the solution, sodium polytungstate powder was dissolved in deionized water by stirring, where its concentration determines both density and viscosity[68]. In all experiments, the depth of this liquid solution was approximately 10 mm, greater than twice the particle diameter. This enabled the particles to move under an adjacent particle without any stiffening by the resistance from the base of the container. However, we did not observe any significant vertical motion of hydrogel particles, potentially due to their low elastic modulus.

To rotate the cylinder and measure the torque, we used a B-type viscometer (BROOKFIELD, LVDV-II+Pro, maximum torque of $6.73 \times 10^{-5}$ N m and rotation speed of $6 \times 10^{-2}$–$1.2 \times 10^3$ deg s$^{-1}$). The motor of the viscometer was set to rotate unidirectionally at a constant rate $\omega$, which was not directly measured. The cylinder and the motor were connected through a torsion spring with a rotational stiffness of $K$ = $9.75 \times 10^{-7}$ N m deg$^{-1}$.

To record and track the motion of all particles in situ, the cylindrical container has a transparent base plane. We used a USB vision camera (OMRON SENTECH, STC-MCCM401U3V, effective pixels of 2048 × 2048, maximum frame rate of 89 fps) with a machine vision lens (RICOH OPTOWL, FL-BC1220-9M). For technical convenience, we used a mirror inclined at 45 deg and set the camera horizontally, with the optical axis aligned to the system center (Fig. 1a). When recording, we placed two panel light sources on both sides of the apparatus to enhance the contrast (Fig. 1b).

In the experimental procedure, we first filled the container with the liquid, added particles, manually stirred the mixture thoroughly, and subsequently inserted the cylinder of the viscometer. After adjusting the height of the glued particles to match that of the particle layer, we started rotating the cylinder while logging torque and recording images.

## Experimental conditions

The shear experiments were performed in 23 runs, as summarized in Supplementary Table 1. We investigated the dependence of mechanical behaviors on porosity (0.18 $\lesssim \phi$ < 0.41 and the corresponding number of hydrogel particles approximately 2700–3900), particle type ($d$ = 4.4 mm hydrogel particles, 4.1 mm glass beads, and 2 mm glass beads), liquid viscosity ($2 \times 10^{-2}$ and 1 Pa s), container diameter (100 and 300 mm), and rotation angular velocity ($\omega$ = 0.06–60 deg s$^{-1}$). For investigating the effect of the particle number, we only varied the total number within a container. The range of porosity was chosen to ensure that the granular layer can maintain a single-layer thickness and also exhibit any detectable torque resistance. To investigate the effect of particle arrangement, we further manually arranged an ordered polycrystalline structure with the intention in runs #67 and #70, where the size of each crystalline cluster was on the order of 10$d$ (Fig. 3c). The particle arrangement in runs #85–87 with the rigid glass beads was also an ordered structure. We also prepared a random structure using a mixture of manually dispersed 3781 hydrogel particles ($d$ = 4.4 mm) and approximately 3000 glass beads ($d$ = 2 mm) in run #77.

The results were obtained at a constant rotation rate of 0.60 deg s$^{-1}$, unless otherwise specified in each figure or dataset. This rotation rate corresponds to a strain rate of $10^{-2}$ s$^{-1}$ at the surface of the rotating cylinder. The rotation rate of 0.60 deg s$^{-1}$ is considered sufficiently slow to neglect the fluid inertial effect with a particle Reynolds number of $Re_p \simeq 0.1$. This value was calculated assuming a viscosity of $2 \times 10^{-2}$ Pa s, a velocity of 0.1 mm s$^{-1}$, a fluid density of 2.8 g cm$^{-3}$, and a particle diameter of 4 mm, respectively.

We estimated the porosity of the granular layer using the mean particle diameter $d$ and total number counted using the recorded image (Fig. 1b). This value represents the bulk porosity and is used as the porosity value in this study. The standard deviation of porosity is 0.008, estimated from that of the hydrogel particle diameter (0.1 mm). We also evaluated whether the estimated porosity was consistent with the visually measured value. In particular, in the case of run #73 using approximately 4000 particles, many particles were out of the monolayer plane. The porosity was estimated to be between 0.15 (assuming all particles are confined within the monolayer) and 0.18 (in run #67 with fewer particles). We provisionally plotted these data in Fig. 4 and Supplementary Fig. 3 at $\phi$ = 0.15. Confining all particles within a single layer proved difficult at lower porosities ($\phi$ < 0.3), leading to some particles being located out of the plane.

## Measurements

We measured the torque and recorded the images immediately after starting the rotational shear for hundreds to tens of thousands of seconds, depending on the runs. The torque was measured at a sampling rate of approximately 2 Hz. With the viscometer, the nominal lower limit of the effective measurement range was $6.73 \times 10^{-7}$ N m (1% of the maximum). However, in practice, both the detection limit and resolution of the system were as low as $7 \times 10^{-8}$ N m (0.1% of the maximum). We recorded the visual images with an exposure time of 0.1 s at a sampling rate of 10 fps. In our experimental setup, a unit pixel corresponds to approximately 0.16 mm at the center of the images. For all images, we did not apply any correction for optical refraction, distortion, and misalignment, which did not qualitatively affect the results of this study. The distortion between the center and edge of the 300 mm container was 2% at maximum, and the shear bands we analyzed

were localized to the central part of the images. In each run, torque measurement and image acquisition were initiated manually at the same time. The accuracy of temporal synchronization between the mechanical data and the visual image was approximately 0.1 s, evaluated by the recording test at 10 fps. Examples of the recorded dataset are shown in Supplementary Fig. 1.

## Analysis

We defined the run time by excluding data at the beginning and final 100 s of each run, and quantified both the mechanical and visual data over the remaining run duration. These exceptional data possibly include unsteady transient flow and noise owing to vibrations and perturbations by the operation of the viscometer.

First, using the mechanical data, we quantitatively characterized the temporal fluctuation of torque $\Gamma(t)$. Torque drop duration $T$ and drop amplitude $\Delta\Gamma = - (\Gamma(t_1 + T) - \Gamma(t_1)) > 0$ with start time $t_1$ are characterized for each event. The values $\Delta\Gamma = \Delta\Gamma^i$, $T = T^i$, $t_1 = t_1^i$ for the i-th event are defined such that $\dot{\Gamma}(t) < 0$ is always satisfied in the range of $t_1^i \leq t < t_1^i + T^i$ and $\dot{\Gamma}(t) \geq 0$ at $t = t_1^i - dt$, $t_1^i + T^i$ with a sampling interval of $dt \simeq 0.5$ s (inset of Fig. 1c).

Although we did not set any threshold value for the definition of the magnitude of torque and its drop amplitude, the events with $\Delta\Gamma \leq 7 \times 10^{-8}$ N m (the practically measurable minimum value) were excluded from the analysis. We analyzed approximately 10–6500 torque drop events, depending on the run duration, as presented in Supplementary Table 1. Event counts are normalized to the duration of each experimental run.

We calculated the cumulative frequency $N(\Delta\Gamma)$, defined as the total number of torque drop events per second with amplitudes equal to or greater than $\Delta\Gamma$. As the cumulative frequency distribution follows a nearly exponential function (Supplementary Fig. 4a for torque drop, Fig. 2a for moment), we fit the following function to the data,

$$N(\Delta\Gamma) = N_0 \exp\left(-\frac{\Delta\Gamma}{G}\right), \tag{3}$$

where $N_0$ and $G$ are the fitting parameters. When an ideal exponential distribution is satisfied, $G = \langle\Delta\Gamma\rangle$, where $\langle\rangle$ denotes the mean value. We obtained the mean torque drop amplitude as the value of fitting parameter $G$.

From the torque data, we also estimated the rotating cylinder displacement $u(t)$ along the side surface of the cylinder. To obtain the displacement, we applied the correction for the apparatus stiffness to the torque data[32]. Using the torsional stiffness of the viscometer $K = 9.75 \times 10^{-7}$ (N m deg$^{-1}$), constant angular velocity $\omega$ (deg s$^{-1}$) of the motor, and radius of the rotating cylinder $R$, tangential displacement at the side surface of the cylinder can be calculated as,

$$u(t) = \frac{K\omega t - \Gamma(t)}{360 K / 2\pi R}. \tag{4}$$

$u(t)$ can be considered the displacement imposed on the granular layer itself, deforming it elastically and plastically. The resolution of the displacement is 0.01 mm, estimated based on that of the torque measurement.

Second, using the recorded raw images of particle arrangements, we obtained three types of images using Fiji/ImageJ software[69]: original binary, time-lapse, and differential images. We first obtained original binary images based on the contrast difference between the particle areas and the background. To obtain them, the original video was converted to 8-bit grayscale. Then, background subtraction was performed using the "Subtract Background" function (100-pixel radius) in Fiji/ImageJ, followed by automatic thresholding using the "Default" method. We used raw images recorded over a 1000 s interval taken more than 1000 s after the beginning of the experiments, which capture the steady arrangements of particles. As an exception,

we used images captured immediately after the beginning of run #67 and 600 s after the beginning of runs #74 and #80 to obtain adequate amounts of recorded images. These exceptional data possibly include unsteady transient flow and noise owing to vibrations and perturbations by the operation of the viscometer. Using the original binary images, we measured the number of particles (Fig. 1b, Supplementary Table 1).

Two types of image processing were applied to the original binary images: time-lapse and differential images (Fig. 3). For the time-lapse images (Fig. 3a–c), we mapped the standard deviation for each pixel over the entire duration of 1000 s. Using the time-lapse images, we characterized the spatial distribution of shear bands (Fig. 3a–c) and measured their thickness (Fig. 4a). To define the shear band thickness $w$, we applied three iterations of erosion followed by three iterations of dilation to the time-lapse images, and then computed the radial distribution of pixel intensity (standard deviation). This processing allows us to extract regions of active particle motion. We empirically identified the shear band edge as the radial position slightly outside the active zone of particle motion with a normalized standard deviation of 250/255, considering the smearing out of the active zone.

For the differential images, we also calculated the absolute difference in brightness between binary snapshots, holding a 2-second gap every 0.1 s (10 fps). We chose the relatively long time gap (2 s) to capture adequate differences between frames as several pixels in each differential image. To reduce the random optical noise, we further applied a Gaussian blur with a standard deviation of 1.00 to the images, and performed a second thresholding in the range 70–255 using the Li method in Fiji/ImageJ. Using the differential images thus obtained, we quantitatively analyzed the motion area where particles have swept (Fig. 3d, e, Supplementary Fig. 1). In addition, to investigate the apparent slip plane size for each event, we also estimated the number of drifting particles $p$ at each time. To calculate $p$, we used the ratio of the differential area over the entire region, $a$, to that per particle over the region with a radius of $r = R + d$ inscribing the fixed particles on the cylinder, $a_c$, resulting in $p = a/a_c$. This value $p$ can be used as an estimate of the equivalent slip plane area $S$. We statistically analyzed $p$ averaged over each duration $T$ for each event (Supplementary Fig. 6). Note that we did not apply static ($\omega = 0$) noise subtraction to the differential images due to difficulties in quantifying the noise. Even with correction, however, $p$ decreased by a constant, causing only a parallel shift in Supplementary Fig. 6 without any qualitative influence on the results.

## Estimation of the moment

Moment $M_0$ of slip associated with stress drop $\Delta\sigma$ is represented as,

$$M_0 = \mu SD = \frac{LS}{C}\Delta\sigma, \tag{5}$$

where $\mu$ is the shear modulus, $S$ is the slip area, $D$ is the slip displacement, $L$ is the slip patch width, $\Delta\sigma$ is the stress drop, and $C$ is the geometrical constant[47]. The stress drop is represented as[47],

$$\Delta\sigma = C\mu\frac{D}{L}. \tag{6}$$

Considering the cylindrical shape of a shear band with stress applied to its outer boundary plane, as shown in the inset of Fig. 4a, the shear stress drop $\Delta\sigma(r)$ with radial distance $r$ from the center of the rotation axis is related to the torque drop amplitude $\Delta\Gamma(r)$ as,

$$\Delta\sigma(r) = \frac{\Delta\Gamma(r)}{2\pi r^2 \delta}, \tag{7}$$

with the diameter $\delta$ of the interparticle contact area, and $L = \delta$, $S = 2\pi r \delta$ under the assumption that the moment arm is mechanically homogeneous. The measured torque drop amplitude on the cylinder $\Delta\Gamma$ can be considered in equilibrium with the torque drop amplitude on the outer boundary plane of the shear band $\Delta\Gamma(r = R + w)$, with radius of the rotating cylinder $R$ and shear band thickness $w$. Using Equations (5) and (7), we can obtain Equation (1). We assumed the following values: $C = 7\pi/16$ for the circular crack at each contact patch[47], and constant $\delta = 0.1$ mm, which is expected to increase with decreasing porosity in practice.

**Model fitting of the moment and moment rate**

Moment $M_0(\Delta\Gamma)$ in Equation (1) is represented using torque drop amplitude $\Delta\Gamma$ as,

$$\Delta\Gamma = (R + w)[2\pi(R + w)\delta]\Delta\sigma, \tag{8}$$

when considering the cylindrical shear band. The mean moment data $\langle M_0 \rangle(\phi)$ in Fig. 4b, obtained from the exponential fitting of the cumulative frequency distribution (Fig. 2a), are adequately explained when we use the $w(\phi)$ relationship and fit $\langle\Delta\sigma\rangle = A\phi^{-n}$ to the mean torque drop amplitude data $\langle\Delta\Gamma\rangle(\phi)$, except for run #73 (solid curve in Supplementary Fig. 3b). The fitting results are $A = 4.7 \times 10^{-13}$ Pa and $n = 5.0$, shown as a solid curve in Fig. 4b.

Moment rate $\dot{M}_0 = M_0/T$ is represented as,

$$\dot{M}_0 = \frac{M_0}{D/\overline{V}}, \tag{9}$$

where $D$ is the tangential slip displacement and $\overline{V} = D/T$ is the mean tangential slip velocity, both at the cylinder surface. As $D$ is calculated from displacement $u(\Gamma)$ in Equation (4), we obtain Equation (2), where $\Delta\Gamma/T$ is trivially expressed with a single variable $\overline{V}$. The analytical curve of Equation (9) using Equation (1) and (2) is shown in Fig. 4c as a solid line, with a constant mean value of $\overline{V} = 0.13$ mm s$^{-1}$ (Supplementary Figs. 3d, 11) and $w(\phi)$ relationship (Fig. 4a).

## Data availability

All data generated in this study are provided in the Source Data file in Zenodo with the identifier doi:10.5281/zenodo.17188918[70]. Image analysis was performed using Fiji/ImageJ, Version 2.14.0/1.54f software[69] in https://imagej.net/software/fiji/ as open source (GNU GPL).

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

## Acknowledgements

We thank O. Kuwano and F. Nakai for their comments and R. Ando for suggesting the perspective of slow earthquakes. We thank Editage (www.editage.jp) for English language editing. Y. S. discloses support for the research of this work from the Japan Science Society [the Sasakawa Scientific Research Grant 2024-6006]. H. K. discloses support for the research of this work from JSPS [KAKENHI Grant Numbers JP23H04134, JP24H00196].

## Author contributions

Y.S. and H.K. conceptualized this experimental study, administered and managed the project, and reviewed and edited the drafts. Y.S. performed investigations, designed the experimental apparatus, established the measurement methodology, curated and formally analyzed data using software coding, visualized data into figures, and wrote an original draft. H.K. supervised this project, provided and installed the experimental equipment, and validated the data and results.

## Competing interests

The authors declare no competing interests.
