## [Transparent Peer Review file · Nature Communications]

Origin of slow earthquake statistics in low-friction soft granular shear

Corresponding Author: Mr Yuto Sasaki

Version 0:

Reviewer comments:

Reviewer #1

(Remarks to the Author)

The manuscript titled “Origin of slow earthquake statistics in low-friction soft granular shear” by Yuto Sasaki and Hiroaki Katsuragi offers compelling experimental evidence for the role of low friction between viscoelastic particles in reproducing two key features of slow earthquakes: exponential frequency-moment distribution and linear scaling between moment and duration. Thanks to an interesting variation of the classical Couette cell, the authors are able to demonstrate how the two aforementioned micro- to meso-scale properties (particle viscoelasticity and interparticle friction) of their experiment can be tuned (by using hydrogel particles or glass beads) to alter the shape of the frequency distribution of torque drops and the scaling between torque drop and duration. This is an interesting demonstration of the spectrum of frictional slip behaviors and how the latter are controlled by system properties. As such, in my view, the insights into the physics governing slow slip will be of interest to the slow-earthquake community.

The authors use a variation of a well-established methodology (see-through Couette cell, with particle tracking) to produce results that support their main argument sufficiently. To my understanding there is no obvious flaw in their methods and discussion of their results. The manuscript contains sufficient information to allow replication of the work, at least in principle. The data supporting the manuscript are accessible and I confirm that they can be accessed and viewed (plotted with Matlab).

To better couple the manuscript with previous, laboratory-based studies looking into the physics of (slow) earthquakes, I would recommend that the authors compare their findings with those of Leeman et al. (2016). For example, can a comparison be made between the role of effective stiffness of a frictional system comprising of hard particles, in the context of rate- and state-dependent friction, and the effect of lubrication in a system with soft particles? This could come as an extension to the comment in line 254 to 257. Another study that I think could be discussed is on the lubricating effect of a fluid on a granular medium, by Reber et al. (2014). A brief discussion of the effect of fluid viscosity and packing fraction on the macroscopic strength of the material could be illuminating. Also, the summarizing Fig. 3 from Reber et al. (2014) could be discussed in the context of Fig. 5 from this manuscript. Finally, Hayman et al. (2011) reported an evolution in the frequency distribution of slip from power law to exponential with increasing packing fraction, in a quasi-2D system of hard particles. The authors could consider updating their discussion in, for example, line 157 to 159 and line 200 to 206, taking into account, e.g., Fig. 3 and Fig. 6 from Hayman et al. (2011).

Finally, some questions, comments and corrections:

Line 9: by “limited” do the authors mean in terms of maximum magnitude?

Line 17: The authors could consider a bridging sentence that would make the transition from earthquakes to hydrogel particle layers smoother.

Line 47: Perhaps it would be simpler to say “... the difference between slow and regular, large earthquakes which control...”? For sure “underground” seems unnecessary.

Line 54: Change “stick” to “stick-“ as in stick-slip.

Line 64, 65: I am not sure what a precursory slip movement looks like in extended data Fig. 1. Perhaps some indicative guiding lines or a blow-up of one example could help.

Line 67, 77/78/79: consider adding an inset (e.g., showing a step or gentle undulation) to better demonstrate what is described in the text.

Line 76: consider adding a statement such as “therefore particles do not fail”, as a clarification

Fig. 2: Perhaps “logarithmic scale” is more appropriate than “logarithmic coordinates”?
Line 115: Add reference to Fig. 4B (larger moment at lower porosity)?
Line 129: do the authors refer to the confining pressure applied to the shear band from the particles in the outer region?
Line 138: Should it not be 17.2 s instead of 15.2 based on Fig. 3E?
Line 198: Does “pressure condition” refer to the confining pressure?
Line 224: Should “Fig. 5E” instead point to “Fig. 2B”?
Line 252: Consider “As *with* slow earthquakes”?
Line 281: Change “plughing” to “ploughing”, if that what is indeed meant.
Line 289, 290: Consider simplifying to “... potentially inducing devastating regular earthquakes, such as... ”.
Line 326: Change “polytundstate” to “polytungstate”.
Line 388: Change “lightning” to “lighting”.
Line 503 & 505: If I understood correctly, both references to “Fig. 2B” should be pointing to “Extended data Fig. 2B” (i.e., not the one in the main text).
Line 603, reference #21: Change “nankai” to Nankai.
Line 670, reference # 47: issue with printing of special characters.
p. 31, extended data Fig. 6A, B: change “Nm” to “N m” in the x-axis caption for consistency.
p. 32, extended data Fig. 8B: change “Nm” to “N m” in the x-axis caption for consistency.

Last but not least, I would like to thank the authors for a thought-provoking manuscript that I read with great interest!

References

1. Leeman, J. R., Saffer, D. M., Scuderi, M. M. & Marone, C. Laboratory observations of slow earthquakes and the spectrum of tectonic fault slip modes. *Nat Commun* 7, 11104 (2016).
2. Hayman, N. W., Ducloué, L., Foco, K. L. & Daniels, K. E. Granular Controls on Periodicity of Stick-Slip Events: Kinematics and Force-Chains in an Experimental Fault. *Pure Appl. Geophys.* 168, 2239–2257 (2011).
3. Reber, J. E., Hayman, N. W. & Lavier, L. L. Stick-slip and creep behavior in lubricated granular material: Insights into the brittle-ductile transition. *Geophys. Res. Lett.* 41, 3471–3477 (2014).

Reviewer #2

(Remarks to the Author)

Connecting the deformation of a wet granular system to slow-slip events is intriguing and potentially of wide interest. The experiments are extensive, and the statistical analysis is intriguing.

One of the shortcomings of the current manuscript is a better motivation why these experiments are relevant to investigating slow slip. The setup and materials are obviously quite different and the connection to recorded slow-slip event locations is not obvious. While the statistics is pointing towards that these experiments indeed captures some behavior, it would be beneficial if there would also be other supporting arguments for the choice of experimental setup and material. In addition, experiments have already been conducted with grains embedded in a fluid where the slip was monitored. While these experiments did not include any statistics it might be fair to cite them non the less (e.g., Reber et al., 2014, GRL). I wonder if the paper would profit from some restructuring of the text. The reader has to wait until the very end to get a better idea on how the experiments potentially relate to slow-slip scenarios. Could you strengthen the relevance at the beginning more? How the porosity in the experiments is related to slow-slip systems is not apparent in the present manuscript. The paper would benefit from a clearer elaboration on this point. What would drive a similar change in porosity in subduction zones?

Some details of the experimental setup that are quite important (when you use a uniform grain size versus a bimodal grain size) are only introduced in line 148. This makes it hard to appreciate this detail in the results presented previously.

Experiment setup: I assume you are keeping the volume of the experimental ‘box’ the same and vary the number of beads to achieve different porosities. Is this correct? If so, how do you control for changes in the local porosity? I assume that the beads can move around in the fluid and that the porosity will not be equal in the entire experimental setup. Does this matter? Some elaboration on this point would be helpful. What would be possible scenarios where you would expect that the porosity is changing similarly to what the experiments are simulating?

Reviewer #3

(Remarks to the Author)

This study elucidates one of micromechanical origins of the scaling laws ubiquitous in slow earthquakes. There are many studies on the scaling laws of regular earthquakes, but those for slow earthquakes are limited. I thus appreciate this study as one of early attempts. An important finding of this paper is that soil mechanical effects could be an explanation for comprehensive slow earthquake scaling laws. It is in contrast with many previous studies that had focused on the frictional properties of real contact areas, rather than granular rearrangements inducing gouge contact reconfigurations. Therefore, I recommend publishing this paper as a proposal of an alternative view on the observed laws of earthquakes.

However, given the great novelty of their proposal, my serious concern is that their contribution to this field is not given across to potential readers in the current format. I am inclined to offer a major revision then.

My concern is mostly about writing, but the lack of references is the most harmful deficit for this work. Introduction rewriting and result comparisons to previous studies are necessary. These comparisons would largely modify the current structure of this paper, so I am not willing to discuss the details.

On introduction-----

Insufficient references on laboratory studies. At least I know several studies already conducted, for example:

0. Laboratory slow earthquakes: Kaproth & Marone 2013; Leeman Marone Safer, 2018

1. Scaling laws of laboratory slow earthquakes: Nielsen et al. 2016 (<http://doi.org/10.1002/2015GL067490>), Wu & McLaskey 2019

2. Gouge effects on the frictional properties: Nakatani 1998; Karner Marone 2001; Chambon Schmittbuhl Corfdir 2002

3. Computationally simulated gouge (I am not familiar with this): e.g. 10.1002/essoar.10504966.1

Referring to the literature 0&1, the previous efforts from laboratories must be outlined. Referring to the literature 2 (&3), the authors should emphasize the significance of gouge effects for frictional properties. I am seriously afraid that this study is overlooked by other experimental research groups without such an effort to connect this study to the literature. As a favorable reviewer of this study, I should offer a thorough restructuring of the introduction section.

Also, I know that there are already other laboratory-based explanations on the slow earthquake scaling laws (cf. the literature 1 above), so I consider this study needs to be read as an alternative view of the observed scaling laws. This point was less written than I had expected. I would like the authors to rewrite it.

Note: here I use granular, gouge, and soil-mechanical effects in a similar meaning. I think the relation between granularity and gouge presence are worth mentioning in the paper, as many readers related to earthquake mechanics are unfamiliar with granular physics. If such jumbling the two is inaccurate to read this paper, additional mentioning is even necessary.

On results-----

Comparisons to the established literature are insufficient in this article. Was this study able to explain some scaling laws unexplained in previous studies? Overlooking Nielsen et al. 2016, Wu & McLaskey 2019 mentioned earlier, scalings themselves seem to be explained in their analysis already. Again, I admit the worth of this paper in proposing another origin of slow-earthquake-like scalings, but even this point is not clear in the current format lacking the references of previous laboratory efforts. In-situ measurements and some other approaches seem advanced but I am not sure. In summary, their section "New implications to slow earthquakes" must be thoroughly rewritten by referring to the previous laboratory efforts. Perhaps the same applies to other sections.

Several questions on the result limitations and details-----

Quasi-two-dimensionality:

The scaling laws could depend on the dimensions of the experimental set up. Is the dimensional limitation of this study unrelated to the results they obtained? The same question applies to monodispersity and sphericity of the system investigated in this study. These points are not well discussed.

Scaling laws vs porosity:

In their results Fig. 2A, the porosity determines the prefactor of the frequency scaling law. Please quantify it (probably another power law?). Is the variety of the frequency prefactor quantitatively consistent with observations? I found qualitative consistency mentioned in their section "New implications to slow earthquakes."

Fluid viscosity, monodispersity, sphericity:

I appreciate their comprehensive confirmation of the result validity, but the link for Extended Data Table 1 and Fig. 5 jumped to Table 1 and Fig. 5 in the paper. The extended data seem not available yet. It may not be the fault of the authors.

I end my comment on this article. This research is obviously backed up with hard work of the authors in laboratories. So I am sure that the results of this paper are worth publishing. At the same time, in reading this article, I felt this paper was written by an early career researcher from a different field of research. It may be fruitful to get advice from laboratory research groups oriented to earthquake mechanics.

Version 1:

Reviewer comments:

Reviewer #1

(Remarks to the Author)

Having studied the revised manuscript as well as the response letter, I believe that the comments I made on the originally submitted manuscript have been adequately addressed. The authors have made a considerable effort to link their findings to published literature, following recommendations from all three reviewers. Compared to the originally submitted version, the revised manuscript offers stronger motivation for the study and improved discussion.

At the discretion of the authors and the associate editor, I recommend that the text be simplified when possible or adjusted for clarity. For example:

- in line 52 of the revised manuscript, I would change "demonstrates analogical behavior similar to the gouge materials" to

“demonstrates behavior analogous to that of gouge”.

- in lines 77 & 78, I would change the wording to “is typically observed in lubricated granular systems”.

- the last sentence of the discussion feels long-winded. Consider a different final clause: “and shear localization that could lead to devastating...”. Additionally, I would replace “pores” in line 310 with “fault zones” or something of similar spatial scale.

Finally, I noticed that the data repository has been updated. I believe there is an issue with the time columns in the updated rawdata.csv. It seems to me that the rounding of the time values is incorrect, leading to several blocks of e.g., torque values having the same time value. Therefore, I would ask the authors to look into this matter.

Reviewer #2

(Remarks to the Author)

The authors addressed my earlier comments in a satisfactory manner. Below are just a couple of small issues I came across while reading the new version.

L 36: Delete: , which is the focus of this study. It disrupts the flow of the paragraph.

L 85: Confusing sentence. Substantially low? Why substantially?

L253: I'm not sure I understand the point you want to make with respect to the experiments only capturing slow slip events in terms of statistics. This seems obvious, as this is the tool you used to investigate the problem.

L255: Somewhat cumbersome first sentence to the paragraph.

Reviewer #3

(Remarks to the Author)

The authors did a great job of revision. They have responded to the questions and comments from the three reviewers through a thorough refinement of the paper. Still, the relation between a thin shear zone in a precut rock and a self-organized shear zone in a fat granular band is unclear, but it is obviously future work. Regarding my comments on the previous version of the manuscript, the current revision addresses the lack of comparison with previous studies and seismological observations, and the paper sectioning has been improved accordingly. I appreciate that several questions I once asked about their results have also been answered. Because the details of the experiments are beyond my expertise, I would like to leave the comments on them to the other two experts. I encourage the publication of this paper in the current format.

Just one unfamiliar wording: wetting conditions -> wet conditions?

A minor confusion (of mine):

The authors describe their lab ruptures as collapses of force chains in a bulk (e.g. L213, L237), but they mention that the main rupture zones are basically near the side wall (above eq.1). The former sounds the observed rupture is a bulk property, whereas the latter sounds the observed rupture is an interfacial property, contributed from the lab machine as well as from their analogue. Which is true? The possibility of the latter makes me consider that their laboratory observations might not derive from a soft material alone but from the interplay of the soft and stiff materials.

Response letter on the revised manuscript titled “Origin of slow earthquake statistics in low-friction soft granular shear”

Dear Editor and Reviewers,

We sincerely appreciate the editor and all three reviewers for your invaluable contributions, as well as your careful, insightful, and positive comments. All reviewers consistently emphasized the importance of clearly positioning our study within the existing literature, recognizing the significance of our findings. In response, we have revised the manuscript with particular attention to the Introduction and Discussion. Additionally, your thoughtful feedback led us to extensively revise the manuscript, and the overall content including manuscript texts and figures has been significantly improved and enriched. As part of the overall revision, we have added three new figures: Supplementary Figures 2, 7, and 8. Within the *Nature Communications* format requirements and word limits, we have incorporated all suggested and additional revisions.

Below, we provide a point-by-point response to each comment. *All received comments are highlighted in brown*, our responses in black, and *excerpts from the revised manuscript in blue along with the line numbers*. For your reference, we also attach a file that shows the differences between the previous version and the revised manuscript. In this file, deletions are indicated in red, and modifications/additions/relocations are highlighted in blue. Please note that line numbers and reference numbers differ between the revised manuscript and the difference file. Therefore, **the line numbers cited in this response letter do not correspond to those in the revised manuscript, but to the difference file. The reference list in this letter matches that of the revised manuscript.**

We sincerely hope our revisions meet the expectations of the editor and reviewers, and we look forward to your favorable consideration of our manuscript.

To Reviewer #1

[1] *The manuscript titled “Origin of slow earthquake statistics in low-friction soft granular shear” by Yuto Sasaki and Hiroaki Katsuragi offers compelling experimental evidence for the role of low friction between viscoelastic particles in reproducing two key features of slow earthquakes: exponential frequency-moment distribution and linear scaling between moment and duration. Thanks to an interesting variation of the classical Couette cell, the authors are able to demonstrate how the two aforementioned micro- to meso-scale properties (particle viscoelasticity and interparticle friction) of their experiment can be tuned (by using hydrogel particles or glass beads) to alter the shape of the frequency distribution of torque drops and the scaling between torque drop and duration. This is an interesting demonstration of the spectrum of frictional slip behaviors and how the latter are controlled by system properties. As such, in my view, the insights into the physics governing slow slip will be of interest to the slow-earthquake community.*

Reply) Thank you for your encouraging comments.

[2] *The authors use a variation of a well-established methodology (see-through Couette cell, with particle tracking) to produce results that support their main argument sufficiently. To my understanding there is no obvious flaw in their methods and discussion of their results. The manuscript contains sufficient information to allow replication of the work, at least in principle. The data supporting the manuscript are accessible and I confirm that they can be accessed and viewed (plotted with Matlab).*

Reply) Thank you for your careful and positive considerations. We sincerely appreciate your effort to access our data and even verify the graph plotting. According to this revision, we updated the open-source data in the Zenodo repository and its reference [70] in the manuscript.

[3] *To better couple the manuscript with previous, laboratory-based studies looking into the physics of (slow) earthquakes, I would recommend that the authors compare their findings with those of Leeman et al. (2016). For example, can a comparison be made between the role of effective stiffness of a frictional system comprising of hard particles, in the context of rate- and state-dependent friction, and the effect of lubrication in a system with soft particles? This could come as an extension to the comment in line 254 to 257.*

Reply) Thank you for your comments and suggestions. We fully agree that examining the effective stiffness in the references (Leeman et al., 2016 [18]; [20]) is of great importance. Reviewer #3 also raised this point, and we kindly ask you to refer to *issue [46.2]* for a comprehensive discussion on this point.

[4] *Another study that I think could be discussed is on the lubricating effect of a fluid on a granular medium, by Reber et al. (2014). A brief discussion of the effect of fluid viscosity and packing fraction on the macroscopic strength of the material could be illuminating. Also, the summarizing Fig. 3 from Reber et al. (2014) could be discussed in the context of Fig. 5 from this manuscript.*

Reply) Thank you very much for pointing us to the relevant literature. We have recognized this as an important point, and Reviewer #2 has also mentioned this in

issue [33]. As you correctly pointed out, the increase in macroscopic strength observed in our study with increasing packing fraction corresponds to the findings and equation (1) in the reference (Reber et al., 2014 [33]), where an increase in fluid viscosity leads to an increase in macroscopic strength (i.e., effective viscosity). In accordance with the effective viscosity increase, the transition from creep to stick-slip occurred in the reference [33] and in ours. Furthermore, the observed localization of the shear band in the presence of fluid, compared to the typical values under dry conditions, is also in agreement with the results reported in fig. 3 from the reference [33] and related studies. We have reflected on these points in the revised manuscript as follows. In addition, as mentioned by Reviewer #2, we have also cited the reference in the Introduction to strengthen the value of our experimental system. The following excerpts show the revised parts we made, highlighted in blue.

Lines 91–93) “This transitional behavior between creep and stick-slip is typically observed in the lubricated granular system [32, 33].”

Lines 155–157) “This shear localization with porosity decrease is also observed in lubricated granular particles with an increase of pore-fluid viscosity [32, 33].”

Lines 366–368) “Our results suggest that with much fluid or ductile matrix at higher porosity, shear localizes within the principal slip zone [33], which is mechanically supported by force chains comprising the rest of the fault core (Fig. 5a).”

Regarding another excerpt, we kindly ask you to also refer to) *issue [32]*

[5] Finally, Hayman et al. (2011) reported an evolution in the frequency distribution of slip from power law to exponential with increasing packing fraction, in a quasi-2D

system of hard particles. The authors could consider updating their discussion in, for example, line 157 to 159 and line 200 to 206, taking into account, e.g., Fig. 3 and Fig. 6 from Hayman et al. (2011).

Reply) We appreciate your insightful comment. As you correctly pointed out, the reference (Hayman et al., 2011 [61]) also reports a power-law distribution with a high exponent (greater than 5), or an exponential distribution, under high packing fraction conditions. We fully recognize that this study is one of the few valuable reports showing such exponential-like statistics. However, while reference [61] also uses a 2D experimental setup, their results contrast with ours: as the packing fraction increases, the event size becomes smaller, and the power-law exponent increases (or the distribution becomes exponential). We believe this discrepancy stems from differences in the experimental setups. Although both our study and reference [61] use a constant-volume condition, their experiment applies simple shear through top and bottom boundaries and, due to practical constraints, also employs sidewalls. As shear deformation proceeds, the aspect ratio of the system decreases, and the hard particles become increasingly constrained by the confining geometry. In other words, despite the nominally constant volume, the effective free volume available for particle motion decreases with strain. Such a setup causes the entire system to behave more elastically, and, as a result, the stress does not reach a steady state, a behavior distinct from what is typically observed in shear experiments such as rotary shear [39] or gouge friction tests. This effect becomes particularly pronounced under high packing fractions, likely leading to the observed suppression of large events in reference [61] (see their figs. 3a, 3b, 6a, and 6b). Therefore, while we value the findings of the

reference [61], a direct comparison with our rotary shear experiments is not straightforward. Although fig. 3c in the reference [61] shows an increase in rigidity and event duration with packing fraction, which may be consistent with our findings, these results likely reflect not only intrinsic properties of the granular system but also effects specific to the simple shear setup. Accordingly, in the final paragraph of the manuscript, we mention this geometrical problem as an important topic for future work, along with a brief reference to the discussion above [61]. We compare our results to other granular experiments of rotary shear [39] and simple shear without aspect-ratio constraints [37] in the revised manuscript, as follows.

Lines 119–120) “The characteristic size (slope) of the exponential size distribution decreases with increasing porosity ϕ , consistent with previous experimental studies [39].”

Lines 125–129) “A slower shear rate results in approaching a linear moment-duration scaling (Supplementary Fig. 5b). An increase in timescale (slow shearing) consistently induces the linear scaling between size and duration in other granular experiments [37] and seismic observations [2], although the underlying mechanism remains unclear.”

Lines 395–397) “As an example of unresolved inconsistency, a simple shear setup with evolving aspect ratio also exhibits exponential size distributions, while its porosity dependence is opposed to ours [61].”

Regarding another excerpt, we kindly ask you to also refer to) issue [47.2]

[6] Finally, some questions, comments and corrections: Line 9: by “limited” do the authors mean in terms of maximum magnitude?

Reply) Thank you for your comments. As you correctly noted, by “limited” we refer to the relatively low maximum event magnitude and the higher relative frequency of smaller events compared to larger ones (a steeper frequency–magnitude distribution). To make this point clearer, we have modified the wording as follows.

Lines 7–8) “Slow earthquakes differ from regular earthquakes in their slower moment release and size distribution dominated by smaller events.”

[7] Line 17: The authors could consider a bridging sentence that would make the transition from earthquakes to hydrogel particle layers smoother.

Reply) Thank you for the suggestion. We have now added a statement regarding the motivation for using hydrogel particles, as follows.

Lines 17–20) “To model slow-earthquake fault materials under hydrothermal conditions, we use a low-friction soft hydrogel particle layer floating on lubricating fluid and conduct stick-slip experiments.”

[8] Line 47: Perhaps it would be simpler to say “... the difference between slow and regular, large earthquakes which control...”? For sure “underground” seems unnecessary.

Reply) Thank you for your comments. We have revised the wording for improved clarity, as follows.

Lines 55–58) “Experimental elucidation of their enigmatic origin would clarify the relationship between slow and regular large earthquakes [2, 24], which control the energy balance of plate tectonics [13].”

[9] Line 54: Change “stick” to “stick-“ as in stick-slip.

Reply) Thank you for your comment. We have revised the wording accordingly as follows.

Lines 77–79) “We captured hundreds of stick- and slow-slip events under quasi-static shear, which become more pronounced with decreasing porosity (Fig. 1c, d)”

[10] Line 64, 65: I am not sure what a precursory slip movement looks like in extended data Fig. 1. Perhaps some indicative guiding lines or a blow-up of one example could help.

Reply) Thank you for your comments. In response to your comments, we carefully re-analyzed the data and found that events with and without precursory slip are indeed mixed. We are sincerely grateful for your insightful feedback. Since preslip does not always occur, we have removed the corresponding description from the manuscript. As you suggested, we have added a magnified view in Supplementary Fig. 1 to better show the relationship between torque and displacement. We have also included guide lines to indicate the onset times of the torque-drop events as defined in our analysis.

Lines 87–89) “The fluid-lubricated soft granular layer under shear exhibits torque fluctuations corresponding to stick–slip events (Fig. 1c, Supplementary Fig. 1).”

Caption for Supplementary Fig. 1) “**Correlation between the torque and particle motion.** The data were obtained in run #83 ($\phi = 0.20$). (a) Torque data. The vertical dashed lines indicate the onset of each event defined by torque drop. (b) Differential area (particle motion area) per second, normalized to the whole area. The vertical dashed lines indicate the onset of each event defined by torque drop. (c) Normalized cross correlation coefficient between the torque and differential area ratio per second, calculated after subtracting the

mean from each time series ($t = 1,890–2,387$ s). The correlation is plotted as a function of the lag time of torque relative to the differential area ratio (particle motion). The peak correlation at slightly negative lag time implies synchronization where the peak of accelerated particle motion follows the corresponding torque drop onset.”

[11] Line 67, 77/ Fig. 1D: consider adding an inset (e.g., showing a step or gentle undulation) to better demonstrate what is described in the text.

Reply) Thank you for your comment. As you suggested, we have added an inset in Fig. 1d for clarity.

Lines 95–97) “A decrease in porosity to approximately 0.20 causes a transition from creep to stick-slip behavior (Fig. 1c, d). The slip rate exceeds the steady shear rate, shown by the dashed black line in the inset of Fig. 1d.”

Caption for Fig. 1d) “(d) Temporal evolution of tangential displacement at the cylinder surface, with color coding corresponding to panel (c). Inset shows an enlarged view of the boxed area. The dashed line represents the steady shear rate of 0.099 mm s^{-1} .”

[12] Line 76: consider adding a statement such as “therefore particles do not fail”, as a clarification

Reply) Thank you for your suggestion. We have revised the expression as follows.

Lines 101-102) “The maximum torque at the minimum porosity is still substantially low, resulting in no particle failure.”

[13] Fig. 2: Perhaps “logarithmic scale” is more appropriate than “logarithmic coordinates”?

Reply) Thank you for your comment. We have corrected the proper wording as follows

(and we have applied the same corrections to the corresponding parts of the other figures as well).

Caption for Fig. 2) “Statistical properties of the released moment. (a) Cumulative frequency distribution of the moment (per second). The linear trends on a semi-log scale demonstrate exponential distributions, with fitting curves shown as solid lines. Inset shows the same data in a log-log plot. The colors and symbols correspond to those in (b). (b) Relationship between moment and duration on a log-log scale for porosities ranging from $\phi = 0.41$ to 0.18.”

[14] Line 115: Add reference to Fig. 4B (larger moment at lower porosity)?

Reply) Thank you for your comment. We have added the reference as follows.

Lines 157–158) “Notably, the released moment exhibits larger values under conditions of lower porosity despite the shear localization (Fig. 4b).”

[15] Line 129: do the authors refer to the confining pressure applied to the shear band from the particles in the outer region?

Reply) Thank you for your comment. Yes, we refer to the radial normal stress. To improve clarity, we have revised the expression as follows.

Lines 170–172) “The outer fixed region can apply low normal stress (radial pressure) to the shear band with higher porosity, constraining its thickness $w(\phi)$.”

[16] Line 138: Should it not be 17.2 s instead of 15.2 based on Fig. 3E?

Reply) We sincerely appreciate your feedback and careful review. We have corrected the numerical value.

Lines 179–180) “Fig. 3e shows the temporal evolution of slip motions over 17.2 s,

corresponding to the torque drops.”

[17] Line 198: Does “pressure condition” refer to the confining pressure?

Reply) Thank you for your comment. As you pointed out, we refer to the normal stress or the confining pressure in 2D corresponding to the various porosities, and this was the intention behind the original wording. However, since the system maintains a constant volume and the pressure is not a trivial constant value, we decided to avoid confusion by mentioning only the control parameter ϕ and have removed references to pressure, as follows.

Lines 237–239) “Thus, the size distribution is governed by the exponentially distributed stress drop $\Delta\sigma$ and its corresponding slip displacement D , which vary with porosity (see Methods; Supplementary Fig. 4).”

[18] Line 224: Should “Fig. 5E” instead point to “Fig. 2B”?

Reply) Thank you for your comment. We have revised the figure references.

[19] Line 252: Consider “As **with** slow earthquakes”?

Reply) Thank you for your comment. Although the corresponding expression no longer appears in the main text due to the overall revision of the manuscript, we have retained a similar expression in the Abstract.

[20] Line 281: Change “plughing” to “ploughing”, if that what is indeed meant.

Reply) Thank you for your comment. The corresponding part has been removed as a result of the overall revision of the manuscript.

[21] Line 289, 290: Consider simplifying to "... potentially inducing devastating regular earthquakes, such as...".

Reply) Thank you for your comment. We have revised the wording for improved clarity as per your suggestion.

[22] Line 326: Change "polytundstate" to "polytungstate".

Reply) Thank you for your comment. We have corrected this word.

[23] Line 388: Change "lightning" to "lighting".

Reply) Thank you for your comment. The corresponding part has been removed as a result of the overall revision of the manuscript.

[24] Line 503 & 505: If I understood correctly, both references to "Fig. 2B" should be pointing to "Extended data Fig. 2B" (i.e., not the one in the main text).

Reply) Thank you for your feedback. We have corrected all the figure references and their hyperlinks, including the mentioned part (please note that the figure numbers have been updated due to the addition of new figures).

[25] Line 603, reference #21: Change "nankai" to Nankai.

Reply) Thank you for your comment. We have corrected the reference list.

[26] Line 670, reference # 47: issue with printing of special characters.

Reply) Thank you for your comment. We have corrected the reference list.

[27] p. 31, extended data Fig. 6A, B: change "Nm" to "N m" in the x-axis caption for consistency.

Reply) Thank you for your comment. We have corrected all the unit formatting in the manuscript, including Supplementary Fig. 10a, b as you mentioned (please note that the figure numbers have been updated due to the addition of new figures).

[28] p. 32, extended data Fig. 8B: change "Nm" to "N m" in the x-axis caption for consistency.

Reply) Thank you for your comment. We have corrected all the unit formatting in the manuscript, including Supplementary Fig. 11b as you mentioned (please note that the figure numbers have been updated due to the addition of new figures).

[29] Last but not least, I would like to thank the authors for a thought-provoking manuscript that I read with great interest!

Reply) We again sincerely appreciate your interest in our work, as well as your encouraging and thoughtful comments and insightful suggestions.

[30] References 1. Leeman, J. R., Saffer, D. M., Scuderi, M. M. & Marone, C. Laboratory observations of slow earthquakes and the spectrum of tectonic fault slip modes. *Nat Commun* 7, 11104 (2016). 2. Hayman, N. W., Ducloué, L., Foco, K. L. & Daniels, K. E. Granular Controls on Periodicity of Stick-Slip Events: Kinematics and Force-Chains in an Experimental Fault. *Pure Appl. Geophys.* 168, 2239–2257 (2011). 3. Reber, J. E., Hayman, N. W. & Lavier, L. L. Stick-slip and creep behavior in lubricated granular material: Insights into the brittle-ductile transition. *Geophys. Res. Lett.* 41, 3471–3477 (2014).

Reply) Thank you for providing the reference information. We now include all

these papers [18, 33, 61] and relevant discussion in the revised manuscript.

To Reviewer #2

[31] Connecting the deformation of a wet granular system to slow-slip events is intriguing and potentially of wide interest. The experiments are extensive, and the statistical analysis is intriguing.

Reply) Thank you for your supportive comments.

[32] One of the shortcomings of the current manuscript is a better motivation why these experiments are relevant to investigating slow slip. The setup and materials are obviously quite different and the connection to recorded slow-slip event locations is not obvious. While the statistics is pointing towards that these experiments indeed captures some behavior, it would be beneficial if there would also be other supporting arguments for the choice of experimental setup and material.

Reply) Thank you very much for your critical and valuable comments. As you pointed out, our experimental setup and conditions differ significantly from the natural fault gouge shear under high temperature and high pressure. However, regarding the experimental materials, hydrogels are particularly suitable for reproducing the low-modulus and lubricated hydrous fault clays, which are metamorphosed under hydrothermal conditions. Using low-modulus materials is also effective to arrest ruptures within the finite-size experimental system [21, 35]. The density and viscosity of the heavy liquid can be adjusted by concentration, which opens the possibility of future experimental studies simulating slow earthquakes in the brittle-ductile transition zone, using this convenient solution as a starting point. In

response to your comments, we have mentioned the necessity and value of our experimental setup and material choices as summarized above. We have revised as follows. In addition, we have revised the related part in the abstract as mentioned by Reviewer #1 (*issue [7]*).

Lines 59–70) “To relate the macroscopic statistics of slow earthquakes to the microscopic dynamics of fault gouge [23, 25–28], a combined approach of stick-slip experiments [29] and in situ fabric observations using soft matter has several advantages. Fault gouge materials generating slow earthquakes are presumed to be rich in low-modulus hydrous clay minerals [30], high-pressure pore fluids [12], and/or viscous phases associated with brittle-ductile transition zones [31]. These factors make a fault low-frictional and soft. Therefore, the mixture of low-friction soft granular material and viscous liquid demonstrates analogical behavior similar to the gouge materials [32–34]. Experiments with soft materials at room temperature and atmospheric pressure allow us to arrest ruptures within the finite system size [21, 35] and track each particle. In such a granular system, porosity not only represents the ratio of fluid or ductile phase, but also controls pressure, fault strength, fluid migration, and slip statistics. Accordingly, this serves as a useful framework for understanding the mechanism of slow earthquake faulting.”

[33] In addition, experiments have already been conducted with grains embedded in a fluid where the slip was monitored. While these experiments did not include any statistics it might be fair to cite them non the less (e.g., Reber et al., 2014, GRL).

Reply) Thank you for your comments. We have come to recognize the significance of this point, and Reviewer #1 also raised

a similar point in *issue [4]*. We have cited relevant studies on deformation experiments of wet granular systems (Reber et al., 2014, GRL [33]; [32]). We kindly ask you to refer to *issues [32, 4]* for the Introduction and the Results/Discussion, respectively.

[34] I wonder if the paper would profit from some restructuring of the text. The reader has to wait until the very end to get a better idea on how the experiments potentially relate to slow-slip scenarios. Could you strengthen the relevance at the beginning more?

Reply) We sincerely appreciate the important feedback and insightful suggestions. Accordingly, we have explicitly clarified the motivation behind connecting our experiments to slow earthquakes in the Introduction. This point is included in *issue [32]*, and therefore we kindly ask you to refer to it regarding the excerpts.

[35] How the porosity in the experiments is related to slow-slip systems is not apparent in the present manuscript. The paper would benefit from a clearer elaboration on this point. What would drive a similar change in porosity in subduction zones?

Reply) We appreciate your critical feedback. We revised the Introduction to emphasize the importance of porosity, which not only represents the depth (pressure) but also controls the gouge strength and pore fluid dynamics in faults. Furthermore, in the Discussion, we pointed out that porosity in slow earthquake faults temporally and spatially varies, controlled by the migration of high-pressure pore fluids, which is one of the driving factors of slow earthquakes. We also cited a seismological observational study implying this phenomenon [60] and discussed its implications. In addition to

the significance of fault-gouge porosity mentioned in the Introduction (*issue [32]*), we have discussed the porosity dependency in the context of the subduction zone system, as follows.

Lines 360–363) “Applying our granular model to fault gouge, the porosity dependence of moment statistics can qualitatively explain the depth distribution of both slow and regular earthquakes along the subducting plate interface [59].”

Lines 379–392) “Furthermore, the dependence of moment on porosity should be observed when local porosity changes according to the pressure of migrating pore fluid. Tectonic tremor events, a type of slow earthquake, exhibit a temporal change in the mean size and occurrence rate, both of which tend to be positively correlated [60]. This correlation is consistent with our results (Fig. 2a, 4b, Supplementary Fig. 2), while the seismological study claims that the size and rate decrease with pore fluid pressure reduction.”

Lines 401–405) “Spatiotemporal observations of slow earthquake statistics could help monitor fluid-filled pores [60], fault strength or stress [24], and shear localization potentially inducing devastating regular earthquakes such as the 2024 M_w 7.5 Noto earthquake [65].”

[36] Some details of the experimental setup that are quite important (when you use a uniform grain size versus a bimodal grain size) are only introduced in line 148. This makes it hard to appreciate this detail in the results presented previously.

Reply) Thank you for your comment. We have added the details of the sample preparation in Methods and Supplementary Table 1, as follows.

Lines 428–442) “We used three types of particles as a fault gouge analogue: opaque

spherical polymer hydrogel particles (Leaf Corporation, Bio Beads, PEG) and spherical glass beads of two sizes. Hydrogel particles have a diameter of $d = 4.4$ mm with a standard deviation of 0.1 mm, with lower friction (coefficient of $< 10^{-1}$) and lower modulus (\sim kPa) [66]. The two types of glass beads have $d = 4.1$ or 2 mm, respectively, with higher friction (coefficient of $\gtrsim 0.2$ [67]) and higher modulus (\sim GPa). To align particle centroids within the same plane on the liquid, we used a single type of monodisperse particle in each experimental run. To investigate the effect of particle arrangement, a bidisperse mixture of the hydrogels and 2 mm glass beads was also used.”

Lines 495–510) “The shear experiments were performed in 23 runs, as summarized in Supplementary Table 1. We investigated the dependence of mechanical behaviors on porosity ($0.18 \lesssim \phi < 0.41$ and the corresponding number of hydrogel particles approximately 2,700–3,900), particle type ($d = 4.4$ mm hydrogel particles, 4.1 mm glass beads, and 2 mm glass beads), liquid viscosity (2×10^{-2} and 1 Pas), container diameter (100 and 300 mm), and rotation angular velocity ($\omega = 0.06$ –60 degs $^{-1}$). For investigating the effect of the particle number, we only varied the total number within a container. The range of porosity was chosen to ensure that the granular layer can maintain single-layer thickness and also exhibit any detectable torque resistance. To investigate the effect of particle arrangement, we further manually arranged an ordered polycrystalline structure with the intention in runs #67 and #70, where the size of each crystalline cluster was on the order of $10d$ (Fig. 3c). The particle arrangement in runs #85–87 with the rigid glass beads was also an ordered structure. We also prepared a

random structure using a mixture of manually dispersed 3781 hydrogel particles ($d = 4.4$ mm) and approximately 3000 glass beads ($d = 2$ mm) in run #77.”

Supplementary Table 1) (please refer).

[37] Experiment setup: I assume you are keeping the volume of the experimental ‘box’ the same and vary the number of beads to achieve different porosities. Is this correct?

Reply) Thank you for your comment. It is exactly correct. We have revised the corresponding part for clarity, as follows.

Lines 75–77) “The experimental system with a constant volume enables the systematic variation of the bulk porosity ϕ and direct tracking of particle motion (Fig. 1b), both of which intrinsically govern the behavior of fluid-retaining faults.”

[38] If so, how do you control for changes in the local porosity? I assume that the beads can move around in the fluid and that the porosity will not be equal in the entire experimental setup. Does this matter? Some elaboration on this point would be helpful.

Reply) Thank you for your thoughtful comment. In this experiment, we simply vary the number of particles dispersed within a fixed system volume, and the porosity is spatially distributed. The shear band in the central region consistently shows a porosity of around 0.3, regardless of the bulk porosity. This value is relatively high compared to the bulk, and its high porosity corresponds to the region where shear localization occurs, since particles are able to move around in the fluid, as you mention. This porosity distribution emerges spontaneously during the experiment and is related to the unresolved problem of shear band self-organization.

Therefore, in this experiment, we are not controlling local porosity but only the

bulk porosity. Although the local porosity within the shear band remains nearly constant under different bulk porosity conditions, the shear band width varies. This suggests that the change in event statistics is due to the difference in shear band width, not the porosity within the shear band. Investigating the influence of local porosity on statistical properties is difficult without first understanding why high-porosity shear bands form spontaneously. This is a fundamentally important problem to be addressed in future work. We have revised the manuscript to clarify these points as follows.

Lines 158–164) “Meanwhile, the porosity within the localized shear band remained constant at 0.3, which is higher (richer in fluid content) than the bulk ϕ , except under the highest ϕ value in run #79 (Supplementary Fig. 3e). Note that we only varied the bulk porosity ϕ ; its spatial distribution spontaneously evolves and stabilizes under steady state.”

Lines 518–520) “We estimated the porosity of the granular layer using the mean particle diameter d and total number counted using the recorded image (Fig. 1b). This value represents the bulk porosity and is used as the porosity value in this study.”

We also kindly ask you to refer to the second excerpt of) *issue [36]*.

[39] What would be possible scenarios where you would expect that the porosity is changing similarly to what the experiments are simulating?

Reply) Thank you for your comments. As the reference [60] observed, porosity in slow earthquake faults could temporally and spatially vary, controlled by the migration of high-pressure pore fluids. Therefore, we expect the porosity-dependent moment

statistics to reflect the pore fluid dynamics within faults. We have revised the manuscript accordingly, and kindly ask you to refer to *issues [32, 35]* regarding the excerpts.

To Reviewer #3

[40] *This study elucidates one of the micromechanical origins of the scaling laws ubiquitous in slow earthquakes. There are many studies on the scaling laws of regular earthquakes, but those for slow earthquakes are limited. I thus appreciate this study as one of early attempts. An important finding of this paper is that soil mechanical effects could be an explanation for comprehensive slow earthquake scaling laws. It is in contrast with many previous studies that had focused on the frictional properties of real contact areas, rather than granular rearrangements inducing gouge contact reconfigurations. Therefore, I recommend publishing this paper as a proposal of an alternative view on the observed laws of earthquakes.*

Reply) We are truly grateful for your thoughtful review and favorable recommendation.

[41] *However, given the great novelty of their proposal, my serious concern is that their contribution to this field is not given across to potential readers in the current format. I am inclined to offer a major revision then. My concern is mostly about writing, but the lack of references is the most harmful deficit for this work. Introduction rewriting and result comparisons to previous studies are necessary. These comparisons would largely modify the current structure of this paper, so I am not willing to discuss the details.*

Reply) We appreciate your critical and insightful comments. In response, we have revised the Introduction, Results, and Discussion to improve their clarity and have incorporated a comparison between our results and previous studies, as detailed below.

[42] *On introduction—
Insufficient references on laboratory studies. At least I know several studies already conducted, for example: 0. Laboratory slow earthquakes: Kaproth & Marone 2013; Leeman Marone Saffer, 2018*

Reply) Thank you for your comments. We have cited previous experimental studies on slow earthquakes to acknowledge their important contributions (Kaproth & Marone, 2013 [16]; Leeman, Marone, & Saffer [20]), as well as other experimental efforts [18, 19, 22]. Reviewer #1 has also raised the relevant reference (Leeman et al., 2016 [18]) in *issue [3]*. For the related revision in both the Introduction and Discussion, we kindly ask you to refer to *issue [46]*.

[43] *1. Scaling laws of laboratory slow earthquakes: Nielsen et al. 2016 (<http://doi.org/10.1002/2015GL067490>), Wu & McLaskey 2019*

Reply) Thank you for your comment. We have also cited the reference (Wu & McLaskey, 2019 [21]) in the main text, acknowledging one of the carefully designed experimental efforts to reproduce slow earthquakes. In Addition, we have cited the suggested reference (Nielsen et al., 2016 [63]) and review paper including comprehensive comparisons between experimental and observational data [64]. For the related revision in both the Introduction and Discussion, we kindly ask you to refer to *issue [46]*.

[44] *2. Gouge effects on the frictional properties: Nakatani 1998; Karner Marone 2001; Chambon Schmittbuhl Corfdir 2002*

Reply) We sincerely appreciate your suggestion of relevant literature. We have added these references (Nakatani, 1998 [25]; Karner & Marone, 2001 [26]; Chambon,

Schmittbuhl, & Corfdir, 2002 [27]). For the related revision in both the Introduction and Discussion, we kindly ask you to refer to *issue* [47].

[45] 3. *Computationally simulated gouge (I am not familiar with this): e.g. 10.1002/essoar.10504966.1*

Reply) Thank you for your critical and wide-ranging suggestions. We have cited the corresponding peer-reviewed publication with the discrete element method (DEM) in the main text (Casas, Mollom, & Daouadji, 2022, Pageoph [62]), while the suggested article is a preprint (Casas, Mollom, & Daouadji, 2020, ESSOAr). For the related revision in both the Introduction and Discussion, we kindly ask you to refer to *issue* [47].

[46] *Referring to the literature 0&1, the previous efforts from laboratories must be outlined.*

Reply) Thank you for your comment. We have acknowledged the previous experimental efforts and added their contributions both in the Introduction and Discussion as follows.

[46.1] In the Introduction, we have provided an overview of previous experimental studies, including those suggested. Most laboratories experimentally reproduce slow earthquakes by reducing the normal stress (i.e., increasing the stiffness normalized with the critical stiffness k/k_c) on the fault plane [16–23]. The normal stress reduction effectively strengthens the normalized stiffness of system, approaching the transitional condition from unstable to stable slip. This transitional slip velocity is slower and the slip duration is longer than highly unstable frictional slip [18, 20]. The velocity dependence of friction in the normalized system

stiffness plays a role in suppressing slip acceleration as well [16, 17]. The reduction of normal stress and the increase in the normalized stiffness also suppress the stressed patch growth. This results in the slow rupture faulting process, that is, slow earthquakes [21], which is consistent with experiments under high pore fluid pressure [22].

However, these references address neither the moment–duration scaling nor size distribution, the scope of our study, which are widely recognized as one of the statistical relationships in slow earthquake observations. One exception is a stick-slip experiment using dehydrating antigorite gouge in the reference [17], where the moment–duration scaling of slow earthquakes is discussed. Given that this paper is most closely related to the main theme of our study, we have decided to include an additional mention of it as a main focus of this study in the Introduction. In accordance with these previous studies, we have revised the manuscript as follows.

Lines 45–55) “Furthermore, in the past decade, many gouge friction experiments have successfully reproduced unstable sliding with slow slip and/or rupture velocities [16–23]. However, the multiple unique statistical laws of slow earthquakes and their origin have scarcely been investigated through experiments, which is the focus of this study. This is due to two main difficulties: reproducing the observed statistics within finite system sizes, and directly observing materials’ internal microstructures during gouge friction experiments. Despite such difficulties, a stick-slip experiment during antigorite serpentinite gouge dehydration successfully demonstrates the roughly linear scaling between stress drop and duration, corresponding to $M_0 \propto T$ [17]. Owing to the lack of experimental data simultaneously reproducing the

multiple statistical properties, the mechanism responsible for the distinct statistical properties of slow earthquakes remains controversial.”

[46.2] In the Results and Discussion, we have added the comparisons with related papers. In the following, we discuss the comparisons unrelated to any scaling relationships [19, 21, 22], for which we kindly ask you to refer to *issue* [51]. As mentioned by Reviewer #1 in *issue* [3], we fully agree that examining the effective stiffness $k' = k/\sigma'_n$ and critical stiffness k_c in slow slip phenomena [18, 20] is of great importance. We addressed the possibility that the observed statistics of slow earthquakes may emerge because the system stiffness k is close to the critical stiffness k_c . However, in our 2D lubricated Couette system, even measuring the radial normal stress σ'_n is a challenging task due to technical issues. It would also require measurements of frictional parameters of $(a - b)$ and D_c under various slip velocity conditions using another high-stiffness apparatus. For these reasons, a direct evaluation of k' and k_c within the framework of the rate-and-state friction law is currently not straightforward. We recognize that elucidating our lubricated system in the framework of the rate-and-state friction remains a highly important issue to be addressed in future studies.

We also note that the characteristics of slow slip are less prominent in the case of hard glass beads (e.g., the frequency distribution of event size deviates from an exponential distribution), which may suggest a decrease in the normalized stiffness k/k_c , though this remains speculative. Alternatively, the presence of lubrication may increase the normalized stiffness k/k_c and enable the emergence of slow slip events as observed in our study. However, as above,

such an interpretation requires a proper evaluation of the stiffnesses k' , k_c . Nevertheless, we have added a statement suggesting that the conditions for the observed slow slip may be close to the stability boundary between stable and unstable slip, as follows.

Lines 340–359) “The mechanism of slow earthquakes proposed by gouge friction experiments is based on frictional instability, which appears to differ from our scenario despite many consistencies. However, the low normal stress condition in our experimental system is consistent with the slow-slip gouge friction experiments characterized by high system stiffness normalized to low critical stiffness [16–18, 20, 21]. When the explanation that transitional slip near the boundary between stable and unstable frictional sliding corresponds to slow slip phenomena [18, 20, 55] is accepted, the stiffness condition in this study possibly lies near the stability boundary. As the critical stiffness reflects the characteristics of rate- and state-dependent friction [56, 57], the persistence of slow slip velocities has been explained within this framework [16, 17, 55, 58]. Accordingly, the bulk frictional properties and relevant porosity evolution of our lubricated soft granular system warrant further investigations in this context.”

[47] Referring to the literature 2 (E3), the authors should emphasize the significance of gouge effects for frictional properties.

Reply) Thank you for your critical feedback. We deeply appreciate your suggestion, which has strengthened our discussion and interpretation. In accordance with your comment, we acknowledge the significance of the force chain development and buckling, i.e., gouge effects of dilation and compaction. We have mentioned the importance of the granularity of gouge material both in

the Introduction and Discussion as follows.

[47.1] In the Introduction, we have revised the manuscript to highlight the importance of the microscopic behavior of particles and force chains within gouge materials in governing macroscopic physical properties such as frictional characteristics. Therefore, we have revised the manuscript and kindly ask you to refer to *issue [32]* regarding the excerpt.

[47.2] In the Results and Discussion, we have added some comparisons with gouge friction experiments, focusing on the effects of force chains and dilatancy. These gouge effects, as you suggested, strongly support our interpretation that force chain dynamics controls the slip statistics within granular materials. The experimental studies [23, 25–28] describe how gouge particles undergo dilation and compaction through the formation and buckling of force chains, leading to shear loading and slip. These results are highly consistent with our experimental results and strengthen our discussion. The reference [27], in particular, reports results consistent with ours by experiments using irregularly shaped, dry quartz sand grains in a similar experimental setup. This study suggests that stick-slip granular dynamics and relevant fabric formation, as we observed, occur regardless of lubrication conditions or particle shape. However, their shear band is wider than that in our results, which supports the interpretation that lubrication promotes the localization of shear bands. We have accordingly added a statement to reflect this point.

The numerical study [62] also discusses the relationship between force chains within the gouge and the bulk loading and stress drop, further supporting the idea that force chains play a key role in controlling slip

behavior. This is consistent with both our interpretation and the results from the gouge experiments above. This study mainly points out that when cohesive forces exist between particles, Riedel shear bands, commonly observed in gouge friction experiments, can form. In our results as well, cohesive forces may be present due to surface tension, and the loosely packed shear band and the radially aligned surrounding vibration region (Fig. 3a–c) might be related to Y-shear and R'-shear types of Riedel shear bands, respectively [40, 62]. Since this remains speculative and given the word limit, we have omitted this discussion but briefly mentioned this topic in the manuscript. Key open question to be explored is how cohesion affects the fabric and the loss of fractality in the contact surface, which we have discussed as a possible cause of the linear moment–duration scaling. Additionally, other numerical simulations of interacting many-body systems suggest that friction modifies the size distribution (Clancy et al., 2006, Burrige-knopoff model: Exploration of dynamic phases, PRE; Saitoh, 2021, The role of friction in statistics and scaling laws of avalanches, EPJE). Although the results of these two studies are important, it is not straightforward to directly compare them with ours, and we could not include these two in the manuscript within the word limit. We have revised the manuscript to better reflect these previous efforts, as follows.

Lines 143–146) “When dry, angular quartz sand grains are sheared in a setup similar to ours, they also exhibit the central shear band and intermittent radial slip propagation into the surrounding region [27]. This implies that the granular fabric we observed is qualitatively consistent, regardless of lubrication and particle shape.”

Lines 153–155) “The shear band thickness of less than ten particle diameters,

which is an approximate typical value for dry frictional conditions [27, 42], is associated with the shorter correlation length of strain under low elastic modulus [43] and low friction.”

Lines 334–339) “Moreover, a critical role of force chain dynamics and its dilatancy/compaction effect has also been recognized by gouge friction experiments [23, 25, 26, 28]. The higher porosity within our localized shear band also implies the dilatancy effect. These studies support our interpretation that force chain development and buckling collapse [45] correspond to the loading and stress drop phases, respectively.”

Lines 393–395) “Building upon our findings, this simple experimental system can facilitate further promising investigations into the effects of pressure in 3D, system geometry [40, 61], granular fabric [23, 28], and rheology [34, 62].”

[48] I am seriously afraid that this study is overlooked by other experimental research groups without such an effort to connect this study to the literature. As a favorable reviewer of this study, I should offer a thorough restructuring of the introduction section.

Reply) We sincerely appreciate your insightful and important feedback. In response, we have revised the manuscript to better position our study within the context of prior research and to highlight comparisons with previous results. In particular, the Introduction has been revised to more clearly delineate the differences from earlier studies and to define the scope of the present work. We kindly ask you to refer to *issues [46, 47, 49, 51]*.

[49] Also, I know that there are already other laboratory-based explanations on the

slow earthquake scaling laws (cf. the literature 1 above), so I consider this study needs to be read as an alternative view of the observed scaling laws. This point was less written than I had expected. I would like the authors to rewrite it.

Reply) Thank you for your comments. In the Introduction, we have revised the manuscript to clarify our position as an alternative explanation of slow earthquakes, compared to the existing experimental studies. Exactly as you pointed out, our study takes a fundamentally different perspective from previous fault gouge friction experiments, in terms of experimental setup and observation target. We aim to capture the essential statistical nature of slow earthquakes using a physically simpler system, which is analogous to a fluid-retaining gouge or a brittle-ductile mixture. We agree that this uniqueness is important. Among the scaling relationships identified in previous gouge friction experiments that have successfully reproduced slow earthquakes [17, 19, 21, 23], the one most relevant to our study is the relationship between stress drop and event duration, experimentally implying the seismological scaling of $M_0 \propto T$ [17]. In the Introduction, these points characterizing our study as an alternative view are clearly mentioned. Thus, we kindly ask you to refer to the excerpts in *issues [32, 46.1]*.

[50] Note: here I use granular, gouge, and soil-mechanical effects in a similar meaning. I think the relation between granularity and gouge presence are worth mentioning in the paper, as many readers related to earthquake mechanics are unfamiliar with granular physics. If such jumbling the two is inaccurate to read this paper, additional mentioning is even necessary.

Reply) Thank you for your comments. We completely agree with your suggestion,

and we recognize that fault gouge can be physically treated as a granular material. In line with your suggestion, we have clarified the two corresponding relationships in the revised manuscript. Since this is related to the necessity and value of our experimental setup, we kindly ask you to refer to *issue [32]* regarding the excerpt.

[51] On results——— Comparisons to the established literature are insufficient in this article. Was this study able to explain some scaling laws unexplained in previous studies? Overlooking Nielsen et al. 2016, Wu & McLaskey 2019 mentioned earlier, scalings themselves seem to be explained in their analysis already. Again, I admit the worth of this paper in proposing another origin of slow-earthquake-like scalings, but even this point is not clear in the current format lacking the references of previous laboratory efforts.

Reply) Thank you for your critical comments. We completely agree on the importance of comparing with previous studies, especially regarding some scaling laws. Besides the scaling relationship captured in our study, roughly five scaling relationships have been identified in previous studies using gouge friction experiments that reproduce slow earthquakes, as described below. Our results, however, provide another scaling relationship obtained from a different perspective along with geophysical observables, distinct from any of the five scalings. Importantly, our experimental results simultaneously realize multiple seismological statistical laws, the size distribution and moment-duration scaling, which have not yet been reported in gouge friction experiments but have been reported in seismological observations. Our experimental system fundamentally differs from previous gouge friction experiments in that it involves

granular materials within a viscous-fluid-dominated matrix, leading to some novel insights into brittle-ductile behavior statistics in this study.

Nevertheless, there are five relationships already observed in lab slow earthquakes: (1) stress drop vs duration, (2) stress drop vs maximum slip velocity, (3) stress drop vs rupture velocity, (4) moment rate spectrum, and (5) fracture energy vs slip displacement.

(1) A negative correlation between stress drop and duration has been commonly observed in gouge friction experiments [19, 21, 23], while ours and antigorite-dehydration-assisted slow slips [17] exhibit the positive scaling. The cause of this inconsistency remains unclear. However, one possible explanation is the presence of abundant fluid in ours and dehydrating gouge, and this point has been added to the manuscript. The fact that our results are consistent with those under dehydration process as slow earthquake faults, is particularly thought-provoking.

(2) The relationship between stress drop and maximum slip velocity is different between slow and regular lab earthquakes [19, 21]. However, depending on experiments, lab slow earthquakes show qualitatively various deviations from the proportional scaling in regular earthquakes, coming from the elastic theory (Brune, 1970, Tectonic stress and the spectra of seismic shear waves from earthquakes, JGR). Furthermore, related geophysical observational data are very sparse, warranting future work in both fields.

(3) The relationship between stress drop and rupture velocity is a positive correlation [21, 22]. As in other gouge experiments, our system is not enough to capture the rupture growth, which we leave for future work.

(4) The moment rate spectrum (moment rate vs frequency), different between regular and slow earthquakes, has been captured by both geophysical observations ([1]; Hawthorne et al., 2018, Observing and modeling the spectrum of a slow slip event, JGR) and by stick-slip experiments [21]. Actually, the statistical differences that we observed should fundamentally result in this spectral difference (Hawthorne et al., 2018). As we did not measure acoustic emissions, this spectral difference is another future work.

(5) Finally, another seismological scaling of fracture energy dissipation as a function of slip displacement is obtained for regular earthquakes, which can be consistently compared between laboratory experiments and geophysical observations [63, 64]. Slow earthquakes are expected to exhibit another characteristic fracture energy scaling law due to viscoelastic dissipation with fluid and ductile phases. However, to our knowledge, no observational or experimental data of the scaling law for fracture energy in the context of slow earthquakes are currently available, even in the relevant references [63, 64]. Thus, we have mentioned this issue briefly in the final paragraph as an important, ongoing issue that should be addressed in future studies, both experimentally and observationally.

In summary, our study investigates the multiple statistical laws in the lab, fundamentally important to geophysical observational results and other scaling laws. In the Discussion, we have refined and limited the scope of our discussion to maintain focus after extensive deliberation within the word limitation, and mentioned the scaling (1), (2), and (5), as follows.

Lines 308–319) “In gouge friction experiments for slow slip phenomena, the linear scaling of stress drop and resultant moment, roughly proportional to duration,

is obtained with dehydrating antigorite serpentinite [17]. This result is similar to ours (Supplementary Fig. 4b). The same study also reports a constant slip velocity, smaller stress drop, and longer duration under dehydration, which are observed in both our system and another viscously lubricated granular shear [32]. These consistently correspond to slow earthquake characteristics and the linear moment-duration scaling of $M_0 \propto T$ in the presence of fluid. In contrast, most gouge friction experiments under conditions ranging from dry to 100% relative humidity show the negative correlation between stress drop and duration across slow to fast slip events [19, 21, 23]. Wetting conditions of the liquid volume confined within pores might play a role in this difference. In addition, note that our experiments only produce slow earthquake equivalents in terms of the statistics.”

Lines 397–399) “Our study lays the groundwork for addressing the scaling between stress drop and slip velocity [19, 21, 22] and the fracture energy scaling of slow earthquakes [63, 64].”

Additional comparisons and discussions beyond these scalings are also provided in) issue [46.2, 47.2]

[52] In-situ measurements and some other approaches seem advanced but I am not sure. In summary, their section “New implications to slow earthquakes” must be thoroughly rewritten by referring to the previous laboratory efforts. Perhaps the same applies to other sections.

Reply) We sincerely appreciate your insightful comments. As far as we have investigated, in-situ tracking of gouge particle motion during friction experiments to probe the microscopic mechanisms remains scarce even today. Most studies approach

the underlying mechanisms through macroscopic mechanical measurements and microstructural observations of gouge materials recovered after the whole experiments. We have thoroughly revised the Discussion section and the manuscript as a whole by incorporating comparisons with previous studies to place our findings in the appropriate context.

[53] Several questions on the result limitations and details——- Quasi-two-dimensionality: The scaling laws could depend on the dimensions of the experimental set up. Is the dimensional limitation of this study unrelated to the results they obtained? The same question applies to monodispersity and sphericity of the system investigated in this study. These points are not well discussed.

Reply) Thank you for your comments.

(1) We consider that dimensionality can affect the results. (2) In contrast, we believe that particle size dispersion does not significantly affect the results qualitatively. (3) As for particle shape, it may influence the results, but it is currently difficult to evaluate, and we regard it as a topic for future investigation. Below, we explain each point in turn.

(1) Regarding dimensionality, while it may not directly influence the size distribution, it may affect the moment-duration scaling. For the size distribution, power-law behavior has been reported even in 2D experiments [36], suggesting that confinement pressure (i.e., normal stress in the radial direction in our system) is likely a more critical parameter than dimensionality. Indeed, since many studies reporting power-law distributions are based on 3D experiments, power-law behavior may more easily emerge under confining pressure in

3D setups. As for moment-duration scaling, there are few prior experimental studies, making it difficult to discuss in detail, but we cannot rule out the influence of dimensionality. We recognize it as an issue that requires further investigation. In actual subduction zones, the depth range of seismicity is limited, which constrains the propagation of fault rupture along the dip direction. In this sense, the quasi-2D shear configuration in our experiments is also physically meaningful in that rupture propagation is restricted in the dip direction. Indeed, there are numerical studies reporting that the moment-duration scalings can be reproduced on finite-width faults [15]. We have revised the manuscript as follows.

Lines 197–217) “First, the exponential size distribution (Fig. 2a) should emerge from the strength of the granular layer, which is generally supported by force chains [44] and induced to slip by their buckling collapse [45]. The stress of both force chains and consisting particles follows an exponential distribution [44, 46]. In contrast to our system, frictional particles under dry conditions and higher normal stress follow power-law size distributions with exponential truncation, regardless of system dimension [36–38]. In the frictional systems under normal stress, the force chains form networks that can cascade into critical slip avalanches. However, lubrication and/or low normal stress (radial pressure) in this study inhibit the formation of force chain network, thereby restricting slip size to the scale of individual particles and suppressing moment release during slip events (Fig. 5a–c). When we substituted hydrogel particles with more rigid glass beads with frictional surfaces, we observed a superposition of exponential and power-law size distributions under lubrication (Supplementary Fig. 10a). These results demonstrate that low friction, as well as

low elastic modulus and low normal stress, plays a crucial role in limiting the mechanically correlated length and preventing cascade collapse events. Consequently, granular systems with these properties manifest the moment statistics that reflect the heterogeneous stress distribution of individual force chains (Fig. 5a, b)."

Lines 261–272) "In contrast, low-modulus particles and/or low normal stress cause the monotonous structures of force chains (Fig. 5a, b) or interparticle patches (Fig. 5e), violating the self-similar scaling. Note that, however, the effects of lubrication and system dimension cannot be ruled out. In particular, the self-similar moment-duration scaling $M_0 \propto T^3$ is not realized in a 2D system due to anisotropic rupture propagation. The fault plane with a finite width, corresponding to the width δ in our granular layer, exhibits a strong dependence of the transition in the moment-duration scaling on stress heterogeneity [15]. Any of these cases potentially lead to a loss of fractality in the distributions of displacement and patch area of slip by the force chain collapse (Supplementary Fig. 4c, 6), resulting in the linear moment-duration scaling $M_0 \propto T$ (Fig. 2b)."

Lines 299–307) "Our results of slow earthquake statistics indicate that lubricated soft particles under low radial pressure and with finite fault width exhibit local force-chain collapse and monotonous structures of interparticle patches. In this context, particles comprising force chains can be considered as gouge grains or even as brittle blocks across multiple scales [50, 51]. Low radial pressure and finite fault width correspond to low effective normal stress due to pore fluid pressure and depth limitation of the seismogenic zone, respectively. These factors mitigate the cascading processes in the on- and off-fault directions, as well as

the self-similarity of faulting, which can be applied to fault gouge materials."

(2) Regarding particle size distribution, we performed experiments using both monodisperse and bidisperse particles and confirmed that the results did not change qualitatively. The particles used in our experiments have a size variation of about 2%, which may prevent them from forming a perfect crystalline arrangement. Since we have revised this point, including the effects of viscosity, monodispersity, and sphericity, we kindly ask you to refer to *issue [55]*, regarding the excerpts.

(3) Finally, regarding particle shape, particle shape and its surface roughness should affect the force chain development, gouge fabric [28], and also event statistics, which also warrants further investigation. At the very least, similar fabrics and particle motions to those in our study have been observed in experiments using sand grains [27], suggesting that our results are consistent in that regard. However, how the statistical laws may change remains nontrivial. As we have discussed in the manuscript, the fractal nature of interparticle contacts could influence statistical properties, and further systematic investigation is needed. Moreover, since gouge materials in subduction zones are typically platy and aligned, it may be necessary to investigate the statistics using such an anisotropic setup as well. Since this point is related to the gouge effects mentioned above, we kindly ask you to refer to the first and last excerpts of *issue [47.2]*.

[54] Scaling laws vs porosity: In their results Fig. 2A, the porosity determines the prefactor of the frequency scaling law. Please quantify it (probably another power law?). Is the variety of the frequency prefactor quantitatively consistent with observations?

I found qualitative consistency mentioned in their section "New implications to slow earthquakes."

Reply) Thank you for your thoughtful comment. We quantified the relationship between the prefactor N_0 (event frequency) and porosity ϕ . Although some runs exhibit a trend that appears to follow a power-law size distribution, as you mentioned, it is not always the case, and we therefore analyze and fit the data using an exponential function. As shown in the newly added Supplementary Fig. 2, the prefactor generally decreases with increasing porosity, indicating that higher porosity leads to a lower event frequency, which is consistent with intuitive expectations. Observational data also show a positive correlation between the average event size (the inverse of the exponential slope) and the overall event occurrence frequency (the exponential prefactor) [60], which is consistent with our experimental findings. However, seismological data are interpreted such that, as fluid pressure decreases over time due to fluid drainage, both the average event size and the event rate decrease [60]. If we interpret the decrease in fluid pressure as a proxy for decreasing porosity, then the porosity dependence observed in the field would be opposite to our experimental results. Since pore pressure is influenced by various factors such as gouge permeability, elastic properties, and fluid pressure or mobility. This implies that a direct correspondence between experiments and observations is not straightforward. Furthermore, natural faults are subjected to confining pressure, resulting in very low porosity conditions. This highlights the need for future experiments under higher normal stress to investigate the effects of low-porosity conditions more thoroughly. We have revised the manuscript as follows.

Lines 98–99) “The frequency of slip events roughly increases with the porosity decrease (Supplementary Fig. 2, Supplementary Table 1).”

Supplementary Fig. 2) (please refer).

We also kindly ask you to refer to the discussion in the second excerpt of) *issue [35]*.

[55] Fluid viscosity, monodispersity, sphericity: I appreciate their comprehensive confirmation of the result validity, but the link for Extended Data Table 1 and Fig. 5 jumped to Table 1 and Fig. 5 in the paper. The extended data seem not available yet. It may not be the fault of the authors.

Reply) Thank you for your comments. We confirmed and corrected all the links throughout the manuscript. We also completed the information in Supplementary Table 1 and added the relevant figures (Supplementary Fig. 7 and 8) as follows.

Lines 189–196) “The low-friction soft granular system exhibits both the exponential size distribution and linear moment-duration scaling (Fig. 2). These two statistical laws remain consistent across various experimental conditions (Supplementary Table 1), including polydispersity (Supplementary Fig. 7), liquid viscosity (Supplementary Fig. 8), and system size (Supplementary Fig. 9). This universality ensures that the two statistical laws can be explained by a simple model, as follows.”

[56] I end my comment on this article. This research is obviously backed up with hard work of the authors in laboratories. So I am sure that the results of this paper are worth publishing. At the same time, in reading this article, I felt this paper was written by an early career researcher from a different field of research. It may be fruitful to get advice

from laboratory research groups oriented to earthquake mechanics.

Reply) We are deeply grateful for your insightful and helpful comments and suggestions. We sincerely apologize for any lack of clarity in our original writing, but your feedback has been invaluable in improving the manuscript, and we believe it is now more accessible to a broader readership. In addition, we have had the opportunity to engage in discussions with researchers working on the experimental mechanics of earthquakes at both domestic and international conferences, including the European Geosciences Union General Assembly 2025 and the Japan Geoscience Union meeting 2025. Insights from those discussions have also been incorporated into the revised manuscript. We hope the revised version meets your expectations and would greatly appreciate your consideration.

References

- [1] Ide, S., Beroza, G. C., Shelly, D. R. & Uchide, T. A scaling law for slow earthquakes. *Nature* **447**, 76–79 (2007).
- [2] Ide, S. & Beroza, G. C. Slow earthquake scaling reconsidered as a boundary between distinct modes of rupture propagation. *Proceedings of the National Academy of Sciences* **120**, e2222102120 (2023).
- [3] Chestler, S. R. & Creager, K. C. Evidence for a scale-limited low-frequency earthquake source process. *Journal of Geophysical Research: Solid Earth* **122**, 3099–3114 (2017).
- [4] Aki, K. Scaling law of seismic spectrum. *Journal of Geophysical Research* **72**, 1217–1231 (1967).
- [5] Kanamori, H. & Anderson, D. L. Theoretical basis of some empirical relations in seismology. *Bulletin of the Seismological Society of America* **65**, 1073–1095 (1975).
- [6] Ishimoto, M. & Iida, K. Observations of earthquakes registered with the microseismograph constructed recently. *Bulletin of the Earthquake Research Institute* **17**, 443–478 (1939).
- [7] Gutenberg, B. & Richter, C. F. Frequency of earthquakes in California. *Bulletin of the Seismological Society of America* **34**, 185–188 (1944).
- [8] Watanabe, T., Hiramatsu, Y. & Obara, K. Scaling relationship between the duration and the amplitude of non-volcanic deep low-frequency tremors. *Geophysical Research Letters* **34**, L07305 (2007).
- [9] Hiramatsu, Y., Watanabe, T. & Obara, K. Deep low-frequency tremors as a proxy for slip monitoring at plate interface. *Geophysical Research Letters* **35**, L13304 (2008).
- [10] Bostock, M. G., Thomas, A. M., Savard, G., Chuang, L. & Rubin, A. M. Magnitudes and moment-duration scaling of low-frequency earthquakes beneath southern Vancouver Island. *Journal of Geophysical Research: Solid Earth* **120**, 6329–6350 (2015).
- [11] Kodaira, S. *et al.* High Pore Fluid Pressure May Cause Silent Slip in the Nankai Trough. *Science* **304**, 1295–1298 (2004).
- [12] Thomas, A. M., Nadeau, R. M. & Bürgmann, R. Tremor-tide correlations and near-lithostatic pore pressure on the deep San Andreas fault. *Nature* **462**, 1048–1051 (2009).
- [13] Araki, E. *et al.* Recurring and triggered slow-slip events near the trench at the Nankai Trough subduction megathrust. *Science* **356**, 1157–1160 (2017).
- [14] Ide, S. & Maury, J. Seismic Moment, Seismic Energy, and Source Duration of Slow Earthquakes: Application of Brownian slow earthquake model to three major subduction zones. *Geophysical Research Letters* **45**, 3059–3067 (2018).
- [15] Weng, H. & Ampuero, J.-P. Integrated rupture mechanics for slow slip events and earthquakes. *Nature Communications* **13**, 7327 (2022).
- [16] Kaproth, B. M. & Marone, C. Slow earthquakes, preseismic velocity

- changes, and the origin of slow frictional stick-slip. *Science* **341**, 1229–1232 (2013).
- [17] Okazaki, K. & Katayama, I. Slow stick slip of antigorite serpentinite under hydrothermal conditions as a possible mechanism for slow earthquakes. *Geophysical Research Letters* **42**, 1099–1104 (2015).
- [18] Leeman, J. R., Saffer, D. M., Scuderi, M. M. & Marone, C. Laboratory observations of slow earthquakes and the spectrum of tectonic fault slip modes. *Nature Communications* **7**, 11104 (2016).
- [19] Scuderi, M. M., Marone, C., Tinti, E., Di Stefano, G. & Collettini, C. Precursory changes in seismic velocity for the spectrum of earthquake failure modes. *Nature Geoscience* **9**, 695–700 (2016).
- [20] Leeman, J. R., Marone, C. & Saffer, D. M. Frictional mechanics of slow earthquakes. *Journal of Geophysical Research: Solid Earth* **123**, 7931–7949 (2018).
- [21] Wu, B. S. & McLaskey, G. C. Contained laboratory earthquakes ranging from slow to fast. *Journal of Geophysical Research: Solid Earth* **124**, 10270–10291 (2019).
- [22] Passelègue, F. X. *et al.* Initial effective stress controls the nature of earthquakes. *Nature Communications* **11**, 5132 (2020).
- [23] Pozzi, G., Scuderi, M. M., Tinti, E., Nazzari, M. & Collettini, C. The role of fault rock fabric in the dynamics of laboratory faults. *Journal of Geophysical Research: Solid Earth* **127**, e2021JB023779 (2022).
- [24] Obara, K. & Kato, A. Connecting slow earthquakes to huge earthquakes. *Science* **353**, 253–257 (2016).
- [25] Nakatani, M. A new mechanism of slip weakening and strength recovery of friction associated with the mechanical consolidation of gouge. *Journal of Geophysical Research: Solid Earth* **103**, 27239–27256 (1998).
- [26] Karner, S. L. & Marone, C. Fractional restrengthening in simulated fault gouge: Effect of shear load perturbations. *Journal of Geophysical Research: Solid Earth* **106**, 19319–19337 (2001).
- [27] Chambon, G., Schmittbuhl, J. & Corfdir, A. Laboratory gouge friction: Seismic-like slip weakening and secondary rate- and state-effects. *Geophysical Research Letters* **29**, 4–1–4–4 (2002).
- [28] Anthony, J. L. & Marone, C. Influence of particle characteristics on granular friction. *Journal of Geophysical Research: Solid Earth* **110** (2005).
- [29] Brace, W. F. & Byerlee, J. D. Stick-Slip as a Mechanism for Earthquakes. *Science* **153**, 990–992 (1966).
- [30] Volpe, G., Collettini, C., Taddeucci, J., Marone, C. & Pozzi, G. Frictional instabilities in clay illuminate the origin of slow earthquakes. *Science Advances* **10**, eadn0869 (2024).
- [31] Behr, W. M. & Bürgmann, R. What’s down there? The structures, materials and environment of deep-seated slow

- slip and tremor. *Philosophical Transactions of the Royal Society A: Mathematical, Physical and Engineering Sciences* **379**, 20200218 (2021).
- [32] Higashi, N. & Sumita, I. Experiments on granular rheology: Effects of particle size and fluid viscosity. *Journal of Geophysical Research: Solid Earth* **114**, 2008JB005999 (2009).
- [33] Reber, J. E., Hayman, N. W. & Lavier, L. L. Stick-slip and creep behavior in lubricated granular material: Insights into the brittle-ductile transition. *Geophysical Research Letters* **41**, 3471–3477 (2014).
- [34] Birren, T. & Reber, J. E. The impact of rheology on the transition from stick-slip to creep in a semibrittle analog. *Journal of Geophysical Research: Solid Earth* **124**, 3144–3154 (2019).
- [35] Dieterich, J. H. Earthquake nucleation on faults with rate-and state-dependent strength. *Tectonophysics* **211**, 115–134 (1992).
- [36] Geller, D. A., Ecke, R. E., Dahmen, K. A. & Backhaus, S. Stick-slip behavior in a continuum-granular experiment. *Physical Review E* **92**, 060201 (2015).
- [37] Abed Zadeh, A., Barés, J., Socolar, J. E. S. & Behringer, R. P. Seismicity in sheared granular matter. *Physical Review E* **99**, 052902 (2019).
- [38] Korkolis, E., Niemeijer, A. R., Paulssen, H. & Trampert, J. A laboratory perspective on the Gutenberg-Richter and characteristic earthquake models. *Journal of Geophysical Research: Solid Earth* **126**, e2021JB021730 (2021).
- [39] Howell, D. W., Behringer, R. P. & Veje, C. T. Fluctuations in granular media. *Chaos: An Interdisciplinary Journal of Nonlinear Science* **9**, 559–572 (1999).
- [40] Daniels, K. E. & Hayman, N. W. Force chains in seismogenic faults visualized with photoelastic granular shear experiments. *Journal of Geophysical Research: Solid Earth* **113**, 2008JB005781 (2008).
- [41] Sibson, R. Thickness of the seismic slip zone. *Bulletin of The Seismological Society of America - BULL SEISMOL SOC AMER* **93**, 1169–1178 (2003).
- [42] Francois, B., Lacombe, F. & Herrmann, H. J. Finite width of shear zones. *Physical Review E* **65**, 031311 (2002).
- [43] da Cruz, F., Emam, S., Prochnow, M., Roux, J.-N. & Chevoir, F. Rheophysics of dense granular materials: Discrete simulation of plane shear flows. *Physical Review E* **72**, 021309 (2005).
- [44] Liu, C.-H. *et al.* Force fluctuations in bead packs. *Science* **269**, 513–515 (1995).
- [45] Tordesillas, A. Force chain buckling, unjamming transitions and shear banding in dense granular assemblies. *Philosophical Magazine* **87**, 4987–5016 (2007).
- [46] Zhang, L., Wang, Y. & Zhang, J. Force-chain distributions in granular systems. *Physical Review E* **89**, 012203 (2014).
- [47] Shearer, P. M. *Introduction to Seismology, Second Edition* (Cambridge Univ. Press, Cambridge, 2009).

- [48] Aharonov, E. & Sparks, D. Rigidity phase transition in granular packings. *Physical Review E* **60**, 6890–6896 (1999).
- [49] Stauffer, D. & Aharony, A. *Introduction To Percolation Theory, Revised Second Edition* (Taylor & Francis, London, 1994).
- [50] Sibson, R. H. Tensile overpressure compartments on low-angle thrust faults. *Earth, Planets and Space* **69**, 113 (2017).
- [51] Sammis, C. G. & Bostock, M. G. A granular jamming model for low-frequency earthquakes. *Journal of Geophysical Research: Solid Earth* **126**, e2021JB021963 (2021).
- [52] Bolton, D. C., Shreedharan, S., Rivière, J. & Marone, C. Frequency-magnitude statistics of laboratory foreshocks vary with shear velocity, fault slip rate, and shear stress. *Journal of Geophysical Research: Solid Earth* **126**, e2021JB022175 (2021).
- [53] Scholz, C. H. The frequency-magnitude relation of microfracturing in rock and its relation to earthquakes. *Bulletin of the Seismological Society of America* **58**, 399–415 (1968).
- [54] Goebel, T. H., Kwiatek, G., Becker, T. W., Brodsky, E. E. & Dresen, G. What allows seismic events to grow big?: Insights from b-value and fault roughness analysis in laboratory stick-slip experiments. *Geology* **45**, 815–818 (2017).
- [55] Baumberger, T., Heslot, F. & Perrin, B. Crossover from creep to inertial motion in friction dynamics. *Nature* **367**, 544–546 (1994).
- [56] Dieterich, J. H. Modeling of rock friction: 1. Experimental results and constitutive equations. *Journal of Geophysical Research: Solid Earth* **84**, 2161–2168 (1979).
- [57] Ruina, A. Slip instability and state variable friction laws. *Journal of Geophysical Research: Solid Earth* **88**, 10359–10370 (1983).
- [58] Segall, P., Rubin, A. M., Bradley, A. M. & Rice, J. R. Dilatant strengthening as a mechanism for slow slip events. *Journal of Geophysical Research: Solid Earth* **115** (2010).
- [59] Obara, K. Characteristic activities of slow earthquakes in Japan. *Proceedings of the Japan Academy, Series B* **96**, 297–315 (2020).
- [60] Nakano, M. & Yabe, S. Changes of event size distribution during episodes of shallow tectonic tremor, Nankai Trough. *Geophysical Research Letters* **48**, e2020GL092011 (2021).
- [61] Hayman, N. W., Ducloué, L., Foco, K. L. & Daniels, K. E. Granular controls on periodicity of stick-slip events: Kinematics and force-chains in an experimental fault. *Pure and Applied Geophysics* **168**, 2239–2257 (2011).
- [62] Casas, N., Mollon, G. & Daouadji, A. DEM Analyses of Cemented Granular Fault Gouges at the Onset of Seismic Sliding: Peak Strength, Development of Shear Zones and Kinematics. *Pure and Applied Geophysics* **179**, 679–707 (2022).

- [63] Nielsen, S. *et al.* Scaling in natural and laboratory earthquakes. *Geophysical Research Letters* **43**, 1504–1510 (2016).
- [64] Cocco, M. *et al.* Fracture Energy and Breakdown Work During Earthquakes. *Annual Review of Earth and Planetary Sciences* **51**, 217–252 (2023).
- [65] Ma, Z. *et al.* Slow rupture in a fluid-rich fault zone initiated the 2024 M_w 7.5 Noto earthquake. *Science* **385**, 866–871 (2024).
- [66] Gong, J. P. Friction and lubrication of hydrogels—its richness and complexity. *Soft Matter* **2**, 544–552 (2006).
- [67] Ishibashi, I., Perry, C. & Agarwal, T. K. Experimental determinations of contact friction for spherical glass particles. *Soils and Foundations* **34**, 79–84 (1994).
- [68] Skipp, G. L. & Brownfield, I. K. Improved density gradient separation techniques using sodium polytungstate and a comparison to the use of other heavy liquids. *Open-File Report (U.S. Geological Survey)* **92–386**, 1–16 (1993).
- [69] Schindelin, J. *et al.* Fiji: an open-source platform for biological-image analysis. *Nature Methods* **9**, 676–682 (2012).
- [70] Sasaki, Y. & Katsuragi, H. Source data for “Origin of slow earthquake statistics in low-friction soft granular shear” Version v2. *Zenodo* <https://doi.org/10.5281/zenodo.16624257> (2025).

Response letter on the 2nd revised manuscript titled “Origin of slow earthquake statistics in low-friction soft granular shear”

Dear Editor and Reviewers,

We sincerely appreciate your invaluable feedback, which has enhanced the significance and value of this paper. Accordingly, we have revised the manuscript, incorporating all the suggested points, editorial requests, and some minor corrections. In addition, we have also made some revisions to improve readability after carefully rereading the manuscript. Below, we provide a point-by-point response to each comment. *All received comments are highlighted in brown*, our responses in black, and excerpts from the revised manuscript in blue along with the line numbers. For your reference, we also attach a file that shows the differences between the previous version and the revised manuscript. In this file, deletions are indicated in red, and modifications/additions/relocations are highlighted in blue. Please note that line numbers differ between the revised manuscript and the difference file. Therefore, **the line numbers cited in this response letter do not correspond to those in the revised manuscript, but to the difference file. The reference numbers in this letter correspond to those in the revised manuscript.**

We sincerely look forward to your favorable consideration of our revision.

To Reviewer #1

[1] *Having studied the revised manuscript as well as the response letter, I believe that the comments I made on the originally submitted manuscript have been adequately addressed. The authors have made a considerable effort to link their findings to published literature, following recommendations from all three reviewers. Compared to the originally submitted version, the revised manuscript offers stronger motivation for the study and improved discussion.*

Reply) Thank you for recognizing the revised points and providing supportive comments.

[2] *At the discretion of the authors and the associate editor, I recommend that the text be simplified when possible or adjusted for clarity. For example: - in line 52 of the revised manuscript, I would change “demonstrates analogical behavior similar to the gouge materials” to “demonstrates behavior analogous to that of gouge”. - in lines 77 & 78, I would change the wording to “is typically observed in lubricated granular systems”. - the last sentence of the discussion feels long-winded. Consider a different final clause: “and shear localization that could lead to devastating...”. Additionally, I would replace “pores” in line 310 with “fault zones” or something of similar spatial scale.*

Reply) Thank you for your comments. In accordance with them, we have revised the manuscript. The following excerpts show the revised parts we made, highlighted in blue.

Lines 51–53) “Therefore, the mixture of low-friction soft granular material and viscous liquid demonstrates behavior analogous to that of gouge [32–34].”

Lines 76–78) “This transitional behavior between creep and stick-slip is typically

observed in lubricated granular systems [32, 33].”

Lines 317–319) “Spatiotemporal observations of slow earthquake statistics could help monitor fluid-filled fault zones [60], fault strength or stress [24], and shear localization that could lead to devastating regular earthquakes such as the 2024 M_w 7.5 Noto earthquake [65].”

[3] *Finally, I noticed that the data repository has been updated. I believe there is an issue with the time columns in the updated rawdata.csv. It seems to me that the rounding of the time values is incorrect, leading to several blocks of e.g., torque values having the same time value. Therefore, I would ask the authors to look into this matter.*

Reply) We really appreciate your critical feedback. We have checked all data files, including the mentioned time values, for rounding errors (to only three significant figures). All CSV files have been regenerated from the original data files. We have uploaded them and updated the citation in the manuscript as [70]. We have confirmed that the measured and calculated values in them retain adequate precision.

To Reviewer #2

[4] *The authors addressed my earlier comments in a satisfactory manner.*

Reply) Thank you for your comments.

[5] *Below are just a couple of small issues I came across while reading the new version. L 36: Delete: , which is the focus of this study. It disrupts the flow of the paragraph. L 85: Confusing sentence. Substantially low? Why substantially? L253: I'm not sure I understand the point you want to make with respect to the experiments only capturing slow slip events in terms of statistics. This seems obvious, as this is the tool you used to investigate the problem. L255: Somewhat cumbersome first sentence to the paragraph.*

Reply) Thank you for your feedback. We have revised the manuscript accordingly as follows.

Lines 35–36) “However, the multiple unique statistical laws of slow earthquakes and their origin have scarcely been investigated through experiments.”

Lines 85–86) “The maximum torque at the minimum porosity remains sufficiently low, resulting in no particle failure.”

Lines 255–260) “In contrast, most gouge friction experiments under conditions ranging from dry to 100% relative humidity and high normal stress show the negative correlation between stress drop and duration across slow to fast slip events [19, 21, 23]. The saturation state of the liquid volume confined within pores, as well as normal stress, might play a role in this difference.”

Lines 261–262) “We could not find any previous work on the size distribution in slow slip gouge friction experiments, as most studies have focused on slip velocity.”

To Reviewer #3

[6] *The authors did a great job of revision. They have responded to the questions and comments from the three reviewers through a thorough refinement of the paper.*

Reply) We appreciate your supportive comments.

[7] *Still, the relation between a thin shear zone in a precut rock and a self-organized shear zone in a fat granular band is unclear, but it is obviously future work.*

Reply) Thank you for your comment. We completely agree with you. Quantitative comparison between fault rock gouge and soft granular media needs some scaling using dimensionless numbers, such as the inertial number [43]. At least, based on our experimental results, porosity reduction localizes the shear band thickness, and fault rock brittle gouge under high normal stress is expected to have a “thin” shear zone as shown in Fig. 5c. In response to this comment, we have revised the manuscript as follows.

Lines 312–315) “Our study lays the groundwork for addressing the scaling between stress drop and slip velocity [19, 21, 22], the fracture energy scaling of slow earthquakes [63, 64], and the modeling of seismic moment as well as shear band thickness [51] through nondimensionalization.”

[8] *Regarding my comments on the previous version of the manuscript, the current revision addresses the lack of comparison with previous studies and seismological observations, and the paper sectioning has been improved accordingly. I appreciate that several questions I once asked about their results have also been answered. Because the details*

of the experiments are beyond my expertise, I would like to leave the comments on them to the other two experts. I encourage the publication of this paper in the current format.

Reply) We are truly grateful for your thoughtful review and encouraging feedback.

[9] *Just one unfamiliar wording: wetting conditions -> wet conditions?*

Reply) Thank you for your comment. We have corrected the wording and kindly ask you to refer to the third excerpt of *issue* [5].

[10] *A minor confusion (of mine): The authors describe their lab ruptures as collapses of force chains in a bulk (e.g. L213, L237), but they mention that the main rupture zones are basically near the side wall (above eq.1). The former sounds the observed rupture is a bulk property, whereas the latter sounds the observed rupture is an interfacial property, contributed from the lab machine as well as from their analogue. Which is true? The possibility of the latter makes me consider that their laboratory observations might not derive from a soft material alone but from the interplay of the soft and stiff materials.*

Reply) We appreciate your critical feedback. We have recognized no wall slip on the side wall of the machine rotary cylinder, which is also roughened with fixed particles. In this paper, we interpret that interparticle displacements within each force chain in the bulk are infinitesimal, and the force-chain buckling explicitly and collectively causes the “slip on the boundary plane within the granular layer between the shear band and surrounding vibration region.” In this granular system, a shear velocity profile exponentially decays in the radial direction, leading to clear discontinuous slip on the outer

lateral surface of the shear band. Therefore, the observed slip statistics derive from the self-organized interplay between the soft shear band (principal slip zone) and the outer stiff granular zone (outer fault core). To avoid misreading, we have replaced the word “outer side wall” with “outer boundary plane” of the shear band. In addition, as we have discussed in the manuscript in response to your previous comments, the system stiffness plays a critical role in slip stability. In accordance with these points, we have revised the manuscript as follows.

Lines 118–119) “The particles principally slip on the boundary plane within the granular layer between the shear band and surrounding vibration region.”

Lines between 183 & 184) “To explain the relationship between the mean moment and porosity, we assume that slips occur repeatedly on the outer boundary plane of the cylindrical shear band, as shown in the inset of Fig. 5a. Although each particle slips within the buckling force chains, we consider that most of the strain energy is dissipated by the slips on the shear-band boundary plane, resulting from collective particle rearrangements within the force chains.”